# Directed differentiation of human iPSCs to functional ovarian granulosa-like cells via transcription factor overexpression

**Merrick D Pierson Smela[1,2†], Christian C Kramme[1,2†], Patrick RJ Fortuna[1,2], Jessica L Adams[1,2], Rui Su[1,2], Edward Dong[1,2], Mutsumi Kobayashi[3], Garyk Brixi[1,2,4,5], Venkata Srikar Kavirayuni[4,5], Emma Tysinger[4,5], Richie E Kohman[1,2], Toshi Shioda[3], Pranam Chatterjee[1,2,4,5], George M Church[1,2]***

[1]Wyss Institute, Harvard University, Boston, United States; [2]Department of Genetics, Harvard Medical School, Boston, United States; [3]Massachusetts General Hospital Center for Cancer Research, Harvard Medical School, Charlestown, United States; [4]Department of Biomedical Engineering, Duke University, Durham, United States; [5]Department of Computer Science, Duke University, Durham, United States

**Abstract** An in vitro model of human ovarian follicles would greatly benefit the study of female reproduction. Ovarian development requires the combination of germ cells and several types of somatic cells. Among these, granulosa cells play a key role in follicle formation and support for oogenesis. Whereas efficient protocols exist for generating human primordial germ cell-like cells (hPGCLCs) from human induced pluripotent stem cells (hiPSCs), a method of generating granulosa cells has been elusive. Here, we report that simultaneous overexpression of two transcription factors (TFs) can direct the differentiation of hiPSCs to granulosa-like cells. We elucidate the regulatory effects of several granulosa-related TFs and establish that overexpression of NR5A1 and either RUNX1 or RUNX2 is sufficient to generate granulosa-like cells. Our granulosa-like cells have transcriptomes similar to human fetal ovarian cells and recapitulate key ovarian phenotypes including follicle formation and steroidogenesis. When aggregated with hPGCLCs, our cells form ovary-like organoids (ovaroids) and support hPGCLC development from the premigratory to the gonadal stage as measured by induction of DAZL expression. This model system will provide unique opportunities for studying human ovarian biology and may enable the development of therapies for female reproductive health.

**\*For correspondence:**
gchurch@genetics.med.harvard.edu

†These authors contributed equally to this work

## Editor's evaluation

This manuscript addresses a fundamental issue in ovarian biology of deriving granulosa cells from human iPS cells. These findings are important to treat female infertility in the future and may prove valuable in the Ob and Gyn clinical practice. The authors provide compelling evidence by developing and validating their model using in vitro ovaroids. This study provides a novel resource for transcriptomic signatures of ovarian somatic cells derived in vitro.

## Introduction

Oogenesis is the central process of female reproduction, yet less is understood about this process in humans relative to other model organisms. This is in part due to difficulties in obtaining samples of human fetal ovaries, and the lack of a suitable model system to study human ovarian development in vitro. The potential power of such a system is illustrated by a recent study in mice, which

**eLife digest** Ovaries are responsible for forming the eggs humans and other mammals need to reproduce. Once mature, the egg cell is released into the fallopian tube where it can be potentially fertilized by a sperm. Despite their crucial role, how eggs are made in the ovary is poorly understood. This is because ovaries are hard to access, making it difficult to conduct experiments on them.

To overcome this, researchers have built artificial ovaries in the laboratory using stem cells from the embryos of mice which can develop into all cell types in the adult body. By culturing these embryonic stem cells under special conditions, researchers can convert them in to the two main cell types of the developing ovary: germ cells which go on to form eggs, and granulosa cells which help eggs grow and mature. The resulting lab-grown ovary can make eggs that produce live mice when fertilized.

This approach has also been applied to human induced pluripotent stem cells (iPSCs), adult human cells which have been reprogrammed to a stem-like state. While this has produced human germ cells, generating human granulosa cells has been more challenging. Here, Pierson Smela, Kramme et al. show that activating a specific set of transcription factors (proteins that switch genes on or off) in iPSCs can make them transition to granulosa cells.

First, the team tested random combinations of 35 transcription factors which, based on previous literature and genetic data, were likely to play a role in the formation of granulosa cells. This led to the identification of a small number of factors that caused the human iPSCs to develop features and carry out roles seen in mature granulosa cells; this includes producing an important reproductive hormone and supporting the maturation of germ cells. Pierson Smela, Kramme et al. found that growing these granulosa-like cells together with germ cells (also generated via iPSCs) resulted in structures similar to ovarian follicles which help eggs develop.

These findings could help researchers build stable systems for studying how granulosa cells behave in human ovaries. This could lead to new insights about reproductive health.

differentiated mouse embryonic stem cells into primordial germ cell-like cells (PGCLCs) and ovarian-like cells, and combined these cell types. Reconstituted follicles were capable of producing oocytes and live offspring (*Yoshino et al., 2021*). Using mouse fetal ovarian somatic cells, a similar system was published that allowed the development of human PGCLCs to the oogonia stage (*Yamashiro et al., 2018*). However, these mixed-species systems are inadequate for fully modeling human ovarian development. Clearly, the construction of fully human ovarian follicles from pluripotent stem cells could enable new advances in the study of female reproduction, epigenetics, and human development.

Ovarian development requires paracrine and juxtacrine interaction between cells from germline and somatic lineages (*Høyer et al., 2005*; *Albertini et al., 2001*; *Rodrigues et al., 2021*; *Su et al., 2004*; *Le Bouffant et al., 2010*; *Otsuka and Shimasaki, 2002*; *Hu et al., 2015*). During embryonic development, the first cells committed to the germline appear at roughly 2–3 weeks post-fertilization (wpf), when primordial germ cells (PGCs) are specified from the posterior epiblast in response to bone morphogenetic protein (BMP) signaling (*Tang et al., 2016*; *Lawson et al., 1999*; *Kee et al., 2006*; *Hayashi et al., 2011*). After specification, PGCs migrate to the developing gonad and undergo further development into oogonia (in female) or spermatogonia (in male) (*Richardson and Lehmann, 2010*; *Grimaldi and Raz, 2020*). The expression of the RNA-binding proteins DAZL and DDX4 (also known as VASA) begins in gonadal PGCs (*Hu et al., 2015*; *Anderson et al., 2007*; *Nicholls et al., 2019*), with DAZL identified as a key factor induced by the gonadal environment that commits PGCs to the germline (*Nicholls et al., 2019*), restricting pluripotency and inhibiting differentiation to somatic lineages (*Chen et al., 2014*). Notably, gonadal PGCs from Carnegie stage 23 primate embryos express mature markers such as DAZL and DDX4, and can complete spermatogenesis upon transplantation into a recipient testis (*Clark et al., 2017*), whereas earlier-stage PGCLCs cannot (*Sosa et al., 2018*).

Previous studies have provided efficient protocols for the differentiation of hiPSCs to DAZL-negative premigratory PGCLCs (*Irie et al., 2015*; *Sasaki et al., 2015*; *Mitsunaga et al., 2017*; *Sebastiano et al., 2021*; *Kee et al., 2006*). However, the only previously reported method that allows human primordial germ cell-like cells (hPGCLCs) to develop to later DAZL-positive stages in vitro is to aggregate them with mouse fetal ovarian (*Yamashiro et al., 2018*) or testicular (*Hwang et al., 2020*; *Kobayashi et al., 2022*) cells, which mimic the environment of the early gonad. Another study tested

aggregation of hPGCLCs with rat neonatal testis cells (*Mall et al., 2020*), but only showed that the hPGCLCs could survive and did not assess maturation.

The somatic cells of the developing ovary provide crucial support for oogenesis. These cells originate from the WT1-positive intermediate mesoderm (*Sasaki et al., 2021*; *Stévant et al., 2019*), and initially exist in a bipotential state, capable of supporting either male or female development. In humans, sex-specific pathways begin to diverge at around 7 wpf (*Hartshorne et al., 2009*). In male mammals, the Y-chromosomal transcription factor (TF) SRY activates a gene regulatory network promoting testis formation and repressing ovary formation (*Koopman et al., 1991*). In the absence of SRY, the TF FOXL2 represses the testis factors, allowing ovarian development to proceed (*Nicol et al., 2018*; *Ottolenghi et al., 2005*). The ovarian somatic cells further differentiate into several lineages, including the granulosa cells which surround and support developing oocytes. Proper oogenesis depends on a complex interplay of paracrine signaling between somatic and germline cells (*Su et al., 2004*; *Le Bouffant et al., 2010*; *Otsuka and Shimasaki, 2002*; *Peng et al., 2013*). Granulosa cells additionally participate in endocrine signaling, producing estradiol and progesterone which regulate diverse reproductive functions (*Findlay et al., 2019*).

Although a few studies have attempted differentiating human pluripotent stem cells to granulosa-like cells through treatment with recombinant signaling proteins, none has resulted in an efficient method, nor has any previous study evaluated their ability to support germ cell development. For example, a study using spontaneous differentiation of hiPSCs produced a small fraction of AMHR2-positive granulosa-like cells which expressed some granulosa marker genes and produced estradiol (*Lipskind et al., 2018*). However, the efficiency of this spontaneous differentiation was low and not numerically reported. A related study used recombinant signaling proteins in a 12-day multistep protocol to direct the differentiation of human embryonic stem cells (hESCs) toward granulosa-like cells (*Lan et al., 2013*); again, the yield was low (12–36%), and although the cells upregulated some marker genes, the expression levels were much lower than observed in primary granulosa cells.

Manipulation of TFs allows reprogramming of cell identity and offers a rapid and efficient way to generate granulosa-like cells. Perhaps the best-known example of cellular reprogramming using TFs is the generation of iPSCs from somatic cells (*Takahashi and Yamanaka, 2006*). TF expression can also guide the differentiation of iPSCs into various somatic lineages, including neurons, fibroblasts, oligodendrocytes, and vascular endothelium (*Ng et al., 2021*). Furthermore, a recent study demonstrated that overexpression of the TFs NR5A1 and GATA4 is sufficient to reprogram human fibroblasts

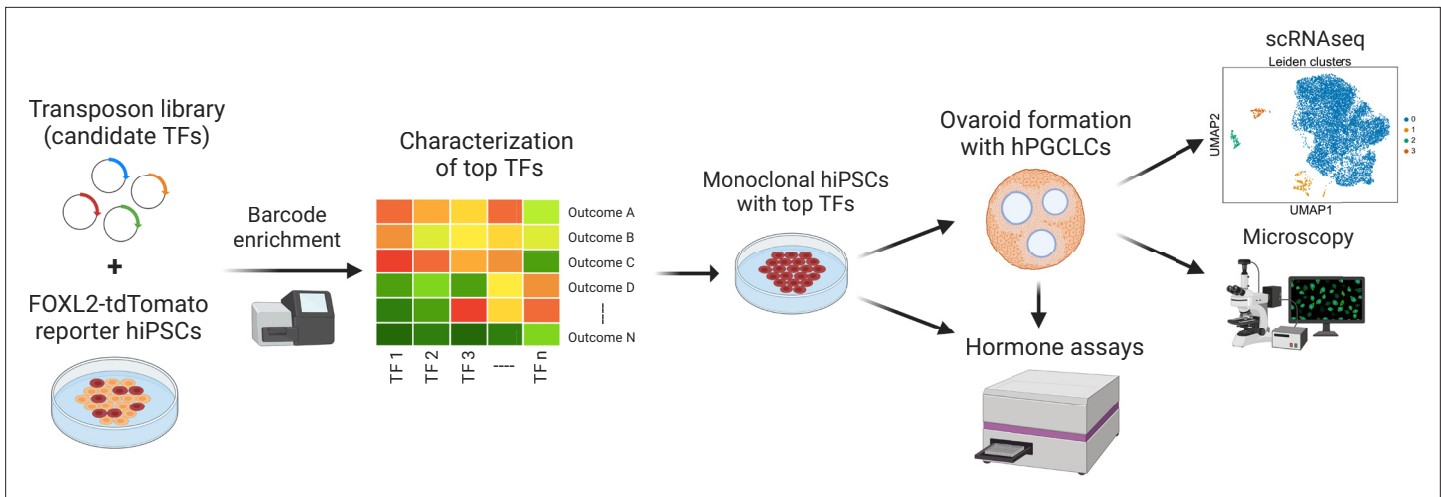

**Figure 1.** Experimental workflow of the study. First, barcoded transcription factor (TF) expression vectors were integrated into FOXL2-T2A-tdTomato reporter human induced pluripotent stem cells (hiPSCs). After induction of TF expression, cells positive for tdTomato and granulosa-related surface markers were sorted, and the barcodes were sequenced. The top TFs based on barcode enrichment were selected for further characterization by combinatorial screening and bulk RNA-seq. Next, monoclonal hiPSC lines were generated that inducibly express the top TFs (see *Figure 3—source data 2*) and generate granulosa-like cells with high efficiency (*Figure 3—figure supplement 2*). Granulosa-like cells from these lines were further evaluated for estradiol production in response to follicle-stimulating hormone (FSH). Finally, they were aggregated with human primordial germ cell-like cells (hPGCLCs) to form ovaroids. These ovaroids produced estradiol and progesterone, formed follicle-like structures, and supported hPGCLC maturation as measured by immunofluorescence microscopy and scRNA-seq.

into Sertoli-like cells (*Liang et al., 2019*), the male developmental counterparts of granulosa-like cells. Therefore, we set out to determine a set of TFs that could be used for an efficient, scalable method of generating granulosa-like cells from hiPSCs. Our overall experimental workflow is shown in *Figure 1*.

## Results

### In silico prediction of granulosa cell-regulating TFs

We began by predicting candidate TFs that could direct the differentiation of hiPSCs to granulosa-like cells. We first selected 21 TFs that were differentially expressed in granulosa cells compared to hESCs and early mesoderm, using previously published datasets for these cell types (*Irie et al., 2015*; *Zhang et al., 2018*). We also included five TFs based on previous developmental biology studies, mainly showing that these TFs were important for mouse ovarian development (*Manuylov et al., 2008*; *Niu and Spradling, 2020*; *Voronina et al., 2007*; *Richards, 2001*; *Nicol et al., 2019*). Finally, we identified nine additional TFs that we predicted to be upstream of the others on the list, based on a gene regulatory network analysis taking into account co-expression data as well as binding motifs (*Kramme et al., 2021a*). The list of TFs (35 in total) is in *Figure 2—source data 2*. Next, we assembled a PiggyBac library by cloning cDNAs into a barcoded destination plasmid (*Kramme et al., 2021b*). The TFs were expressed under the control of a doxycycline-inducible promoter, which we characterized in previous work (*Kramme et al., 2021a*).

### Overexpression screening identifies TFs that drive granulosa-like cell formation

To enable identification of granulosa-like cells, we used CRISPR/Cas9-mediated homology-directed repair to engineer an hiPSC line with a homozygous knock-in of a T2A-tdTomato reporter at the C-terminus of FOXL2 (*Figure 2—figure supplement 1*). We chose FOXL2 due to its specific expression in granulosa cells (*Ottolenghi et al., 2005*; *Cocquet et al., 2002*). Next, the pooled cDNA library was co-electroporated with a PiggyBac transposase expression plasmid, and a population of cells with integrated transposons was selected by treatment with puromycin. In these experiments, we tested several different library compositions and DNA concentrations (see Methods, *Figure 2—source data 2*, and *Figure 2—figure supplement 2*).

For screening, we induced TF expression with doxycycline, and sorted reporter-positive cells after 5 days of treatment. In addition to screening TF expression in the pluripotency-supporting mTeSR Plus medium, we also expressed TFs following differentiation of hiPSCs to early-stage mesoderm by treatment with the GSK3 inhibitor CHIR99021. The TF expression resulted in a small fraction of FOXL2+ cells, from which we extracted gDNA and sequenced barcodes. In both conditions, we observed barcodes for *NR5A1* to be strongly enriched in FOXL2+ cells relative to negative cells, as well as relative to the barcodes in the pre-induced hiPSC population (*Figure 2A*). Other TFs showed more modest barcode enrichment (*RUNX2*, *GATA4*, and *TCF21*), or enrichment in only one condition (*KLF2* and *NR2F2*). Interestingly, barcodes for *FOXL2* were strongly depleted (*Figure 2A*), suggesting negative feedback whereby exogenous FOXL2 directly or indirectly suppresses the expression of the endogenous FOXL2 reporter allele.

Using the top TFs identified in barcode screening (*NR5A1*, *RUNX1/RUNX2*, *TCF21*, and *GATA4*) we then further optimized the conditions for generation of FOXL2+ cells. We included *RUNX1* in this list along with *RUNX2* because the two TFs are structurally and functionally similar, and *RUNX1* is known to play an important role for granulosa cell maintenance in the mouse (*Nicol et al., 2019*). We integrated these TFs into hiPSCs and established monoclonal lines, which we screened by flow cytometry after TF induction. We monitored FOXL2-tdTomato as well as the surface markers CD82, follicle-stimulating hormone receptor (FSHR), and EpCAM. CD82 is absent in hiPSCs, but highly expressed in granulosa cells beginning at the primordial follicle stage (*Zhang et al., 2018*). FSHR is specific for late-stage (secondary/antral) granulosa cells and Sertoli cells (the male equivalent). By contrast, EpCAM is expressed in hiPSCs and epithelial cells, but not granulosa cells (*Zhang et al., 2018*).

Out of 23 lines tested, 4 showed >50% FOXL2-tdTomato+CD82+EpCAM− cells after 5 days of differentiation (*Figure 3—figure supplement 1A*). We observed low levels of FSHR expression, suggesting that we were mainly generating cells corresponding to early (primordial/primary follicle) granulosa cells, given that FSHR only becomes strongly expressed at later stages of ovarian follicle development

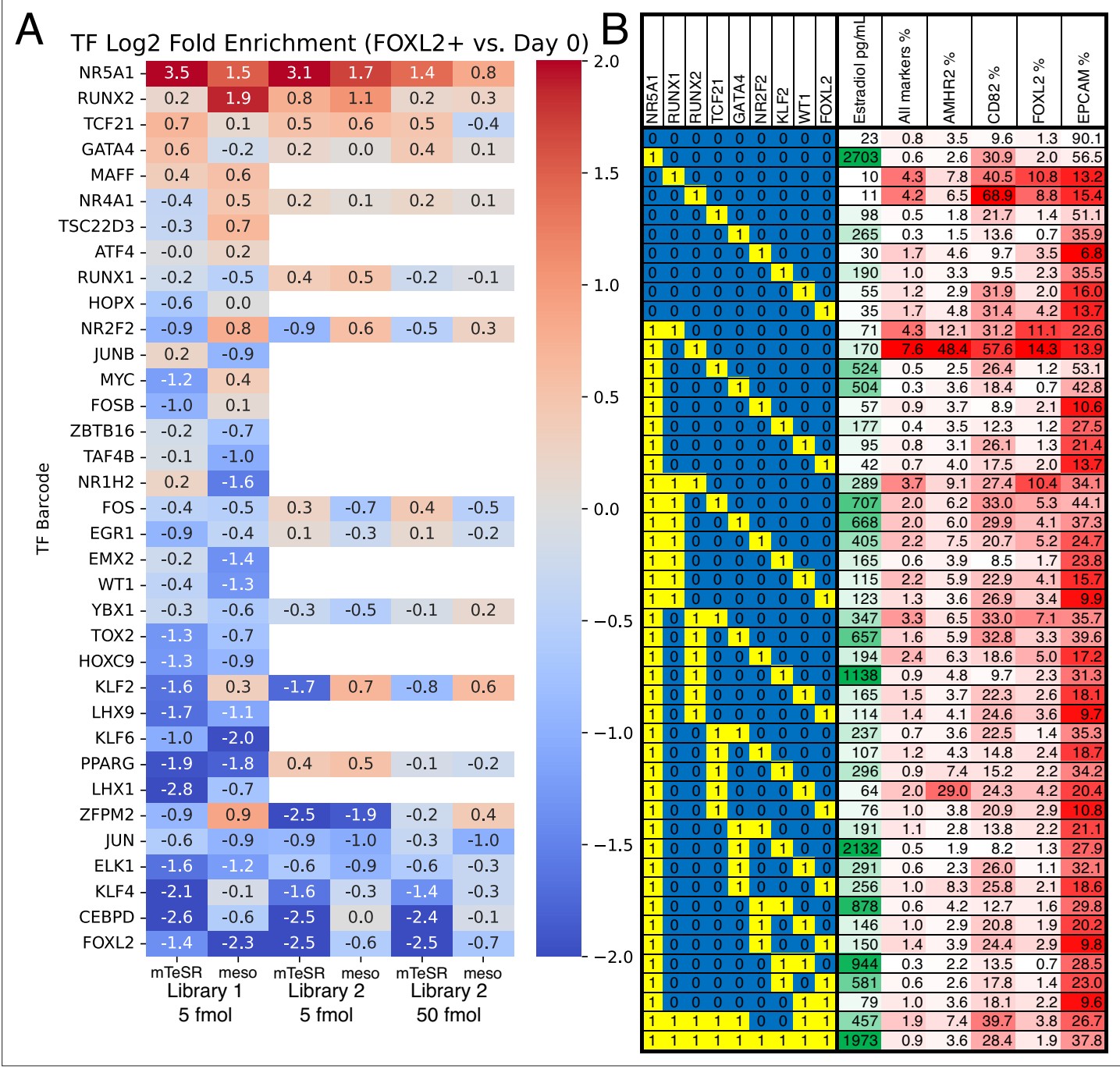

**Figure 2.** Identification of transcription factors (TFs) whose overexpression generates granulosa-like cells. (**A**) Pooled screening of barcoded TF cDNA libraries (see Methods for details) identifies TFs enriched in FOXL2-T2A-tdTomato+ cells. Library #1 is the full library of 35 TFs, and library #2 is a library containing a subset of 18 TFs. Empty values correspond to TFs that were absent in library #2. (**B**) Combinatorial screening identifies minimal TF combinations for inducing granulosa-like cells. TF combinations were integrated into human induced pluripotent stem cells (hiPSCs); a '1' in the left-hand box signifies the presence of the TF in the combination corresponding to that row. For each combination, the polyclonal hiPSC population was differentiated with TF induction (see Methods). For the last 24 hr of differentiation, cells were additionally treated with FSH and androstenedione. Estradiol production and granulosa markers were measured by ELISA and flow cytometry after a total of 5 days. NR5A1 expression induced high levels of estradiol synthesis, but the combination of NR5A1 with RUNX1 or RUNX2 was required to give the best results for granulosa markers. 'All markers' signifies FOXL2+CD82+AMHR2+EPCAM−.

The online version of this article includes the following source data and figure supplement(s) for figure 2:

**Source data 1.** Barcode counts for the enrichment analysis.

*Figure 2 continued on next page*

*Figure 2 continued*

**Source data 2.** List of transcription factors (TFs) included in libraries 1 and 2.

**Figure supplement 1.** Construction of the FOXL2-T2A-tdTomato reporter human induced pluripotent stem cell (hiPSC) line.

**Figure supplement 1—source data 1.** Raw gel scans for FOXL2 reporter genotyping.

**Figure supplement 2.** Control of transcription factor (TF) expression plasmid copy number delivered to human induced pluripotent stem cells (hiPSCs).

**Figure supplement 2—source data 1.** qPCR data for copy number analysis.

---

(*Oktay et al., 1997*). Further optimization by adding the GSK3 inhibitor CHIR99021 for the first 2 days (*Supplementary file 2*) resulted in a near-homogeneous FOXL2-tdTomato+CD82+EPCAM− population for the top cell lines (*Figure 3—figure supplement 2*), with this effect being doxycycline responsive. Genotyping of two top lines (*Figure 3—figure supplement 1C*) revealed that both had *NR5A1* expression cassettes integrated, with the first line also having *RUNX1* and the second line *RUNX2*.

Subsequently, we set out to determine which TFs were sufficient for induction of granulosa-like cells. We tested combinations of hit TFs from our screening (*NR5A1, RUNX2, TCF21, GATA4, KLF2,* and *NR2F2*) as well as TFs reported to be important for ovarian function (*RUNX1, WT1* [−KTS isoform], and *FOXL2*). We tested expression of each TF individually, as well as combinations of other TFs and our top hit *NR5A1*. In addition to flow cytometry, we also measured production of estradiol after treatment of the cells with FSH and androstenedione. We observed that combinations containing *NR5A1* and *RUNX1* or *RUNX2* upregulated FOXL2-tdTomato expression, as well as granulosa surface markers AMHR2 and CD82 (*Figure 2B*). Estradiol production was strongly induced by *NR5A1*, and weakly by *GATA4* and *KLF2. RUNX1* and *RUNX2* expression somewhat decreased estradiol production, suggesting a regulatory role of these factors. Overall, these results indicate that *NR5A1* and either *RUNX1* or *RUNX2* are sufficient to induce a granulosa-like phenotype.

## TF-mediated differentiation drives granulosa-like cell formation based on gene expression signatures

We next examined the gene expression of our granulosa-like cells. We compared the bulk transcriptomes of hiPSCs, COV434 and KGN ovarian tumor cells, and sorted FOXL2+CD82+ granulosa-like cells from day 5 of a polyclonal differentiation with expression of our previously identified top TFs (NR5A1, TCF21, GATA4, and RUNX1; see *Figure 2*). As an additional control, we included hiPSCs differentiated under the same conditions but without TF induction.

In the absence of TF expression, the hiPSCs differentiated into cells expressing mesoderm markers (full gene expression data are provided in the Source Data for *Figure 3*). At day 5 of differentiation, we observed strong upregulation relative to hiPSCs of genes associated with both lateral mesoderm (*HAND1* and *BMP5*) as well as paraxial mesoderm (*PAX3*), which potentially indicates the presence of a heterogeneous population. Upregulated genes were enriched for gene ontology terms related to heart, blood vessel, muscle, and skeletal development (*Supplementary file 3*).

With TF induction, we observed expression of bipotential gonad and granulosa markers, notably including *AMHR2, CD82, FOXL2, FSHR, IGFBP7, KRT19, STAR,* and *WNT4* (*Figure 3A*). The ovarian stromal/theca cell marker *NR2F2* was also upregulated. The expression levels (transcripts per million (TPM)) were generally comparable to those observed in previously published data from granulosa cells and human fetal gonad (*Zhang et al., 2018*; *Sybirna et al., 2020*). The only major exception was *WT1*, which had much weaker expression than in vivo (although still greater than hiPSCs and KGN cells). We also examined the expression of pluripotent and adrenal markers to check for incomplete or off-target differentiation. We did not observe adrenal marker expression, although we did note some residual *POU5F1* (encoding the OCT4 protein) expression that may indicate that our polyclonal population did not fully differentiate. COV434 cells, which were commonly considered as granulosa-like cells but recently reclassified as small cell ovarian carcinoma (*Price et al., 2012*; *Zhang et al., 2000*; *Karnezis et al., 2021*), did not express most granulosa markers, but did express high levels of *WT1* and *NR2F2*. By contrast, KGN cells expressed high levels of most granulosa markers, but not *WT1*.

We also performed a transcriptome-wide comparison of our TF-induced granulosa-like cells with three published datasets spanning weeks 6–21 of human fetal ovarian development (*Tang et al., 2015*; *Leclucze et al., 2020*; *Yatsenko et al., 2019*). Tang et al. sequenced RNA from male and female gonadal somatic cells at gestational week 7, whereas Yatsenko et al. and Leclucze et al. sequenced

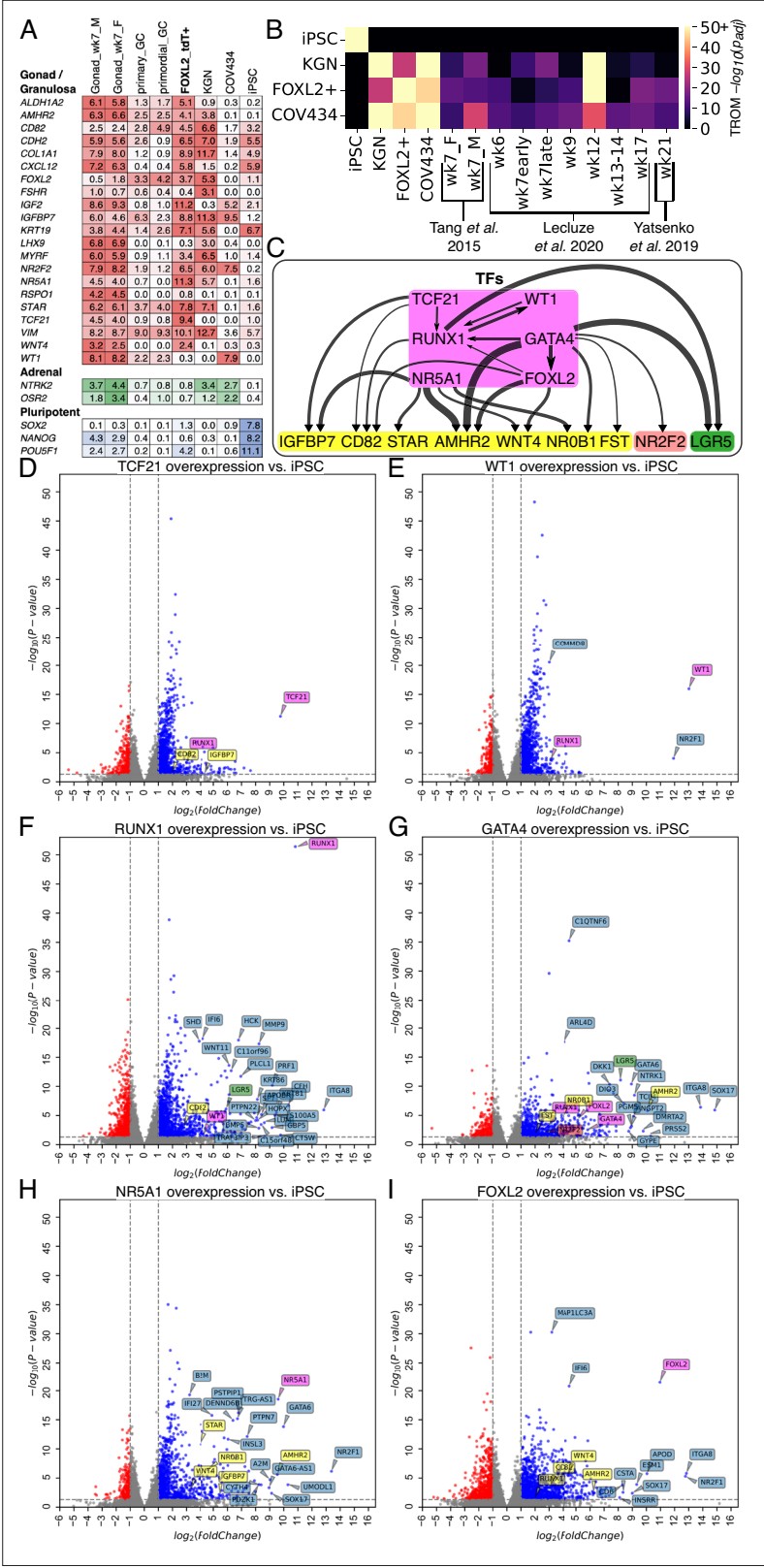

**Figure 3.** Transcriptomic analysis of transcription factor (TF)-induced granulosa-like cells. (**A**) Gene expression of selected markers in granulosa-like cells. Log$_2$(TPM) values for gondal/granulosa, adrenal, and pluripotent marker genes were compared between 7 wpf male and female fetal gonad somatic cells, primary and primordial granulosa cells, TF-induced FOXL2+ cells, KGN cells, COV434 cells, and human induced pluripotent stem cells (hiPSCs).

*Figure 3 continued on next page*

*Figure 3 continued*

(**B**) Transcriptome overlap measure (TROM) comparison of TF-induced FOXL2+ cells, COV434 cells, and hiPSCs with published in vivo data from different time points in ovarian development. (**C**) Regulatory effects of granulosa-related TFs. RNA-seq was performed after 2 days of TF overexpression in hiPSCs (TFs shown in magenta). A differential gene expression (DEG) analysis was performed for all samples relative to the hiPSC control ($n = 2$ biological replicates each). Black arrows represent significant (false discovery rate <0.05) upregulation, with the width proportional to the $\log_2$-fold change. Interactions are shown between TFs (magenta) and granulosa marker genes (yellow), as well as the stromal/theca marker *NR2F2* (red) and the pre-granulosa marker *LGR5* (green). (**D–I**) Volcano plots showing DEGs in the TF overexpression experiments. Colors are as in panel C; other DEGs not listed in panel C are shown in blue. Not all DEGs could be labeled due to space limits, but they are listed in the Source Data for this figure.

The online version of this article includes the following source data and figure supplement(s) for figure 3:

**Source data 1.** Gene expression data from bulk RNA-seq.

**Source data 2.** List of monoclonal human induced pluripotent stem cell (hiPSC) lines used for granulosa-like cell production.

**Figure supplement 1.** Evaluation of monoclonal human induced pluripotent stem cell (hiPSC) lines for yield and quality of granulosa-like cells.

**Figure supplement 1—source data 1.** Flow cytometry data and raw gel scans.

**Figure supplement 2.** Monoclonal human induced pluripotent stem cell (hiPSC) lines with integrated transcription factors (TFs) allow the efficient production of granulosa-like cells in response to doxycycline.

**Figure supplement 2—source data 1.** Flow cytometry data.

RNA from whole fetal ovaries over a range of gestational weeks. Using the transcriptome overlap measure (TROM) (*Li et al., 2017*), we computed similarity scores between our samples and these datasets (*Figure 3B*). We found a high degree of similarity ($-\log_{10}(p_{adj}) = 50.5$) between our granulosa-like cells and human fetal ovaries. The closest match was with gestational week 12 whole fetal ovary from the Leclucze et al. dataset (which sequenced RNA from whole ovaries at different gestational weeks) but we also observed a significant similarity ($-\log_{10}(p_{adj})$ of 12.7–15.8) with the other two datasets we examined. By contrast, COV434 and KGN cells showed a more modest similarity to the fetal ovary, and hiPSCs showed none at all.

To further elucidate the regulatory effects of individual TFs that we identified as granulosa related (FOXL2, WT1, NR5A1, GATA4, TCF21, and RUNX1), we integrated their cDNA plasmids into hiPSCs and induced expression with doxycycline. For these experiments, we harvested RNA after 2 days of TF expression, since we were interested in short-term effects. We performed differential gene expression analysis relative to hiPSCs, and examined regulatory effects among TFs, and between TFs and marker genes (*Figure 3C*). Overall, we observed that FOXL2, NR5A1, GATA4, and RUNX1 had the greatest effects on granulosa marker genes. Therefore, we generated additional hiPSC lines by integrating these TFs (*Figure 3—figure supplement 1*). To rule out any effects of the T2A-tdTomato reporter allele potentially perturbing FOXL2 function, we generated these in a wild-type background (F66 hiPSC line). We measured surface marker expression by flow cytometry after induction, confirmed expression of the granulosa marker *AMHR2* by qPCR, and additionally measured estradiol production in the presence of FSH and androstenedione (*Figure 3—figure supplement 1B*). We selected the clones (*Figure 3—source data 2*) that showed the most robust overall phenotype based on a combination of surface markers, *AMHR2*, expression and estradiol production.

In the differential expression analysis, we additionally found that GATA4 upregulated *NR2F2*, a marker of ovarian stromal and theca cells (*Figure 3C, G*). *LGR5*, previously reported as a marker of pre-granulosa cells (*Niu and Spradling, 2020*), was upregulated by GATA4 and RUNX1 (*Figure 3C, F, G*). We also confirmed a previous report that FOXL2 expression upregulated the cyclin-dependent kinase inhibitor *CDKN1B* (*Gustin et al., 2016*), as well as downregulating *ORC1* and *MYC* (Source Data for *Figure 3*), suggesting suppression of cellular proliferation.

Finally, we conducted a gene ontology enrichment analysis of upregulated and downregulated genes for each TF (*Mi et al., 2021*). Significantly enriched terms were mainly related to generic developmental processes (e.g., 'tissue development', see *Supplementary file 3*) although terms related to gonad development were also significantly enriched for NR5A1 upregulated genes.

## Granulosa-like cells respond to FSH and perform steroidogenesis

We next validated the ability of our granulosa-like cells to carry out one of the key endocrine functions of granulosa cells: the production of estradiol. In the ovary, theca cells convert cholesterol to androstenedione, which is the substrate for estradiol production in granulosa cells. The rate-limiting step is oxidative decarboxylation by CYP19A1 (aromatase), producing estrone, which is subsequently reduced to estradiol by enzymes in the HSD17B family, typically HSD17B1 in granulosa cells (*Hakkarainen et al., 2015*). In vivo, this pathway of estrogen synthesis is stimulated by FSH (*Sasson et al., 2004*; *Welsh et al., 1984*).

We treated our granulosa-like cells with androstenedione, in the presence or absence of FSH or forskolin (which directly increases levels of the FSHR second messenger cAMP). As controls, we used COV434 and KGN ovarian tumor cells, which were previously reported to produce estradiol from androstenedione (*Price et al., 2012*; *Zhang et al., 2000*; *Nishi et al., 2001*), as well as immortalized primary human granulosa cells (HGL5) and primary adult mouse ovarian somatic cells. Our granulosa-like cells produced estradiol from androstenedione, and in seven out of the nine monoclonal lines we tested, this steroidogenic activity significantly increased upon stimulation with FSH or forskolin (*Figure 4A*, see Source Data for statistical tests). One of our granulosa lines (F3/N.T #5) produced high levels of estradiol in all conditions, and this line, unlike the others, had neither *RUNX1* nor *RUNX2* expression vectors integrated (*Figure 3—figure supplement 1C*).

The levels of estradiol produced by our three FSH-responsive lines were similar to those produced by KGN human granulosa tumor cells, which also showed responses to FSH and forskolin (*Figure 4A*). In contrast, COV434 cells, which showed no *FSHR* expression in our RNA-seq data (*Figure 3A*), were unresponsive to FSH alone, producing estradiol only in the presence of forskolin. HGL5 immortalized human granulosa cells did not produce estradiol under any condition. Primary adult mouse ovarian somatic cells produced similar amounts of estradiol to our hiPSC-derived granulosa-like cells (*Figure 4A*). However, the mouse cells did not show a response to FSH or forskolin, possibly because they were already exposed to FSH in vivo.

We also investigated whether our granulosa-like cells maintained their steroidogenic activity during ovaroid (ovarian organoid) co-culture with hPGCLCs (see below). We measured hormone levels in ovaroid supernatants in the presence or absence of androstenedione and FSH. In addition to estradiol, we also measured progesterone, which granulosa cells produce in vivo after ovulation and formation of the corpus luteum. We observed production of both hormones in five out of six samples (*Figure 4B*). Estradiol was produced only in the presence of androstenedione supplementation, and levels increased with FSH treatment. Progesterone was produced in all conditions but was highest in the absence of androstenedione.

## Granulosa-like cells support germ cell development within ovaroids

Current methods for inducing and culturing hPGCLCs produce cells corresponding to immature, premigratory PGCs that lack expression of gonadal PGC markers such as DAZL (*Irie et al., 2015*). During fetal development, PGCs mature through interactions with gonadal somatic cells, with DAZL playing a key role in downregulation of pluripotency factors and commitment to gametogenesis (*Nicholls et al., 2019*; *Chen et al., 2014*). This process has recently been recreated in vitro using mouse fetal ovarian somatic cells (*Yamashiro et al., 2018*), which allowed the development of hPGCLCs to the oogonia-like stage. We hypothesized that in vitro-derived human granulosa-like cells could perform a similar role, with the potential for eliminating interspecies developmental mismatches. Therefore, we combined our granulosa-like cells with hPGCLCs to form ovarian organoids, which we termed ovaroids.

To generate ovaroids, we aggregated these two cell types in low-binding U-bottom wells, followed by transfer to air–liquid interface Transwell culture. As a comparison, we followed a previously described protocol (*Yamashiro et al., 2020*) to isolate fetal mouse ovarian somatic cells and aggregate them with hPGCLCs. By immunofluorescence, we observed expression of the mature marker DAZL beginning in a subset of OCT4 + hPGCLCs at 4 days of co-culture with hiPSC-derived granulosa-like cells (*Figure 5A*). In contrast, robust DAZL expression in co-culture with mouse cells was not observed until day 32 (*Figure 5B*), with fainter expression visible at day 26 (*Supplementary file 4*). Similarly, in a previous study using the same hPGCLC line and anti-DAZL antibody, DAZL expression was observed only after 77 days of co-culture with mouse fetal testis somatic cells (*Kobayashi et al., 2022*).

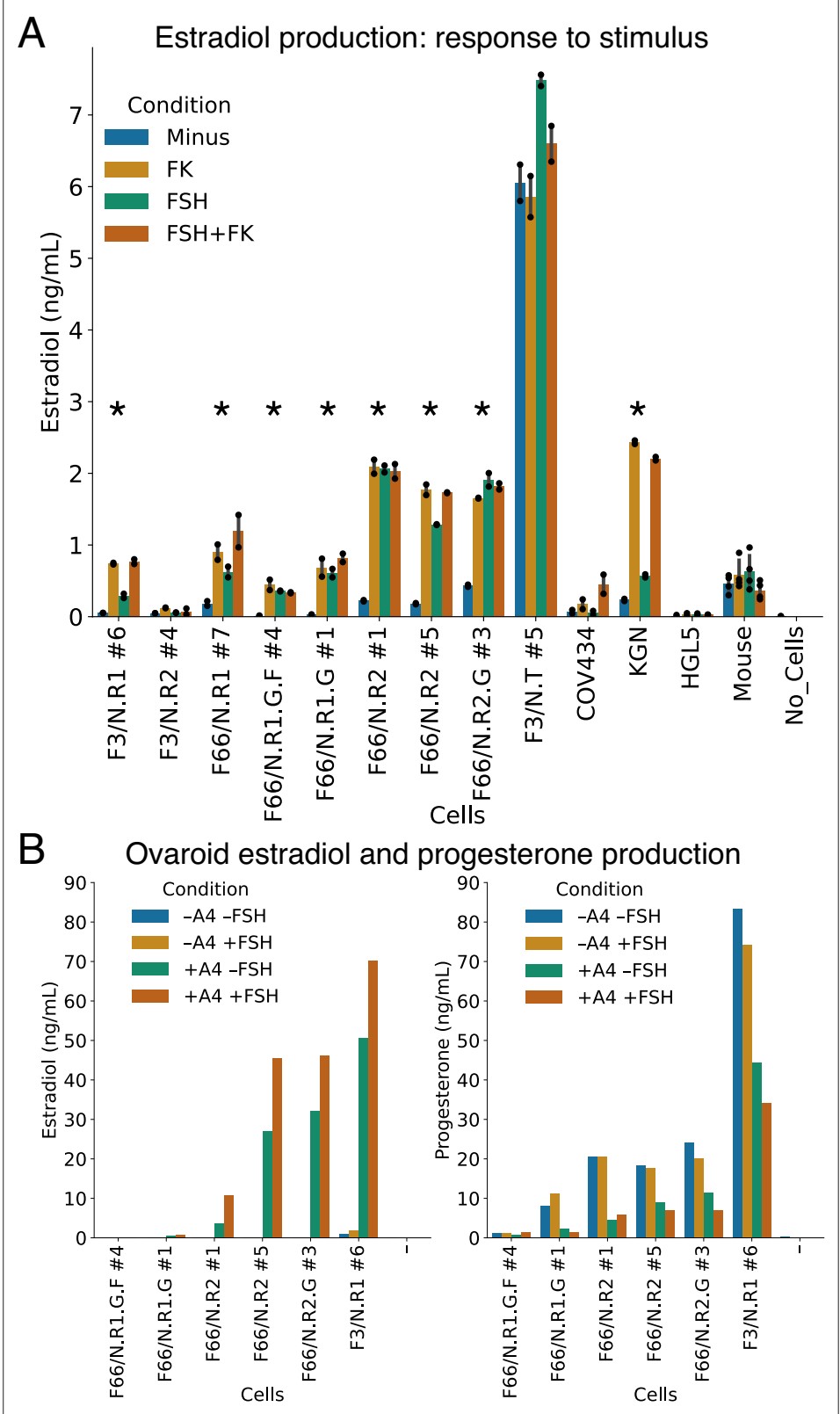

**Figure 4.** Hormonal signaling by granulosa-like cells. (**A**) Granulosa-like cells produce estradiol in the presence of androstenedione and either FSH or forskolin (FK). Results are shown from nine monoclonal populations (see *Figure 3—source data 2*) of granulosa-like cells (*n* = 2 biological replicates for each of 9 clones, error bars are 95% CI), as well as the COV434 and KGN human ovarian cancer cell lines, HGL5 immortalized primary human granulosa

*Figure 4 continued on next page*

*Figure 4 continued*

cells, and primary adult mouse granulosa cells. Asterisks mark lines where FSH production significantly (two-tailed *t*-test, p < 0.05) increased upon stimulation. Exact p values are given in the Source Data. (**B**) Ovaroids produce both estradiol and progesterone. Estradiol production requires androstenedione and is stimulated by FSH. Results are shown for ovaroids formed with six different monoclonal samples of granulosa-like cells (*n* = 1 sample per ovaroid per condition), at 3 days post-aggregation.

The online version of this article includes the following source data for figure 4:

**Source data 1.** Hormone assay data.

The fraction of DAZL+ cells reached its maximum at day 14 in human ovaroids and day 38 in mouse ovaroids (*Figure 5C*). In human ovaroids, the fraction of OCT4+ cells declined after day 8. In mouse ovaroids, the fraction of OCT4+ cells also declined over time. At day 16 in human ovaroids, DAZL+OCT4− cells were also apparent (*Figure 5—figure supplement 1*) in addition to DAZL+OCT4+ cells, and past day 38 there were more DAZL+ cells than OCT4+ cells in total (*Figure 5C*). The down-regulation of OCT4 in DAZL+ oogonia occurs in vivo during the second trimester of human fetal ovarian development (*Anderson et al., 2007*); however, we did not observe the transition of DAZL

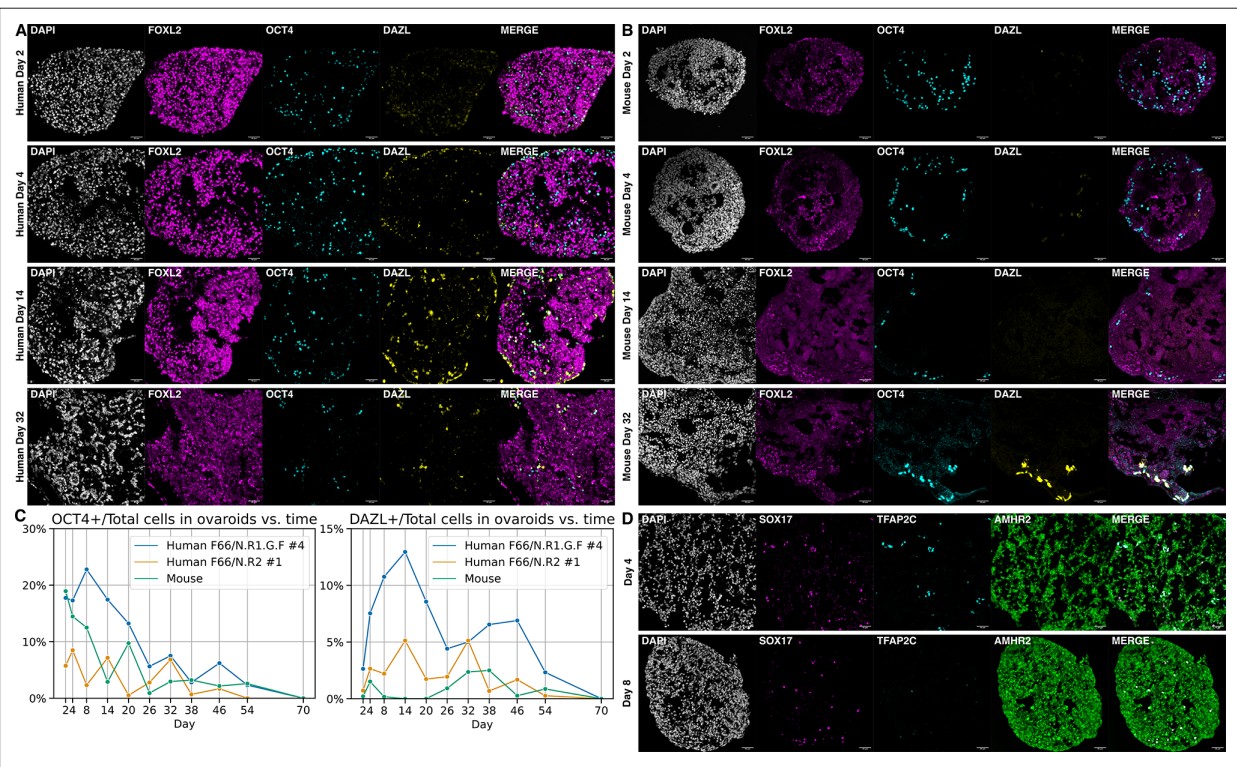

**Figure 5.** Ovaroid development and germ cell maturation. Scale bars in immunofluorescence images are 40 μm. (**A**) Human ovaroid (F66/N.R1.G.F #4 granulosa-like cells + hPGCLCs) sections at days 2, 4, 14, and 32 of culture, stained for FOXL2 (granulosa), OCT4 (germ cell/pluripotent), and DAZL (mature germ cell). (**B**) Mouse ovaroid (fetal mouse ovarian somatic cells + hPGCLCs) sections stained as in panel A. (**C**) Fraction of OCT4+ and DAZL+ cells relative to the total (DAPI+) over time in human ovaroids and mouse xeno-ovaroids. Counts were performed at 11 time points on images from 2 replicates of human ovaroids (F66/N.R1.G.F #4 and F66/N.R2 #1 granulosa-like cells + hPGCLCs) and 1 replicate of mouse xeno-ovaroids. (**D**) Human ovaroid (F66/N.R2 #1 granulosa-like cells + hPGCLCs) sections at days 4 and 8 of culture, stained for SOX17 (germ cell), TFAP2C (early germ cell), and AMHR2 (granulosa).

The online version of this article includes the following source data and figure supplement(s) for figure 5:

**Source data 1.** Ovaroid images (hPGCLC + F66/N.R1.G.F #4).

**Source data 2.** Ovaroid images (hPGCLC + F66/N.R2 #1).

**Source data 3.** Ovaroid images (hPGCLC + mouse fetal ovarian cells).

**Source data 4.** Cell counts from ovaroid images.

**Figure supplement 1.** DAZL and OCT4 expression observed by immunofluorescence in day 16 ovaroids.

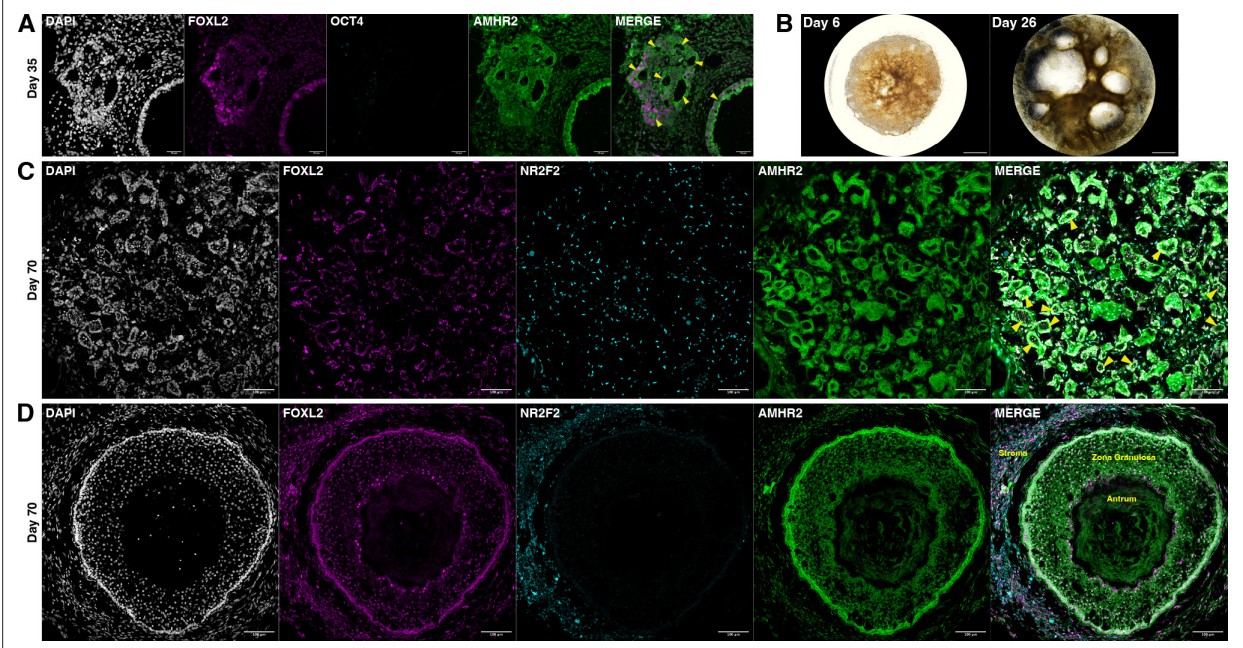

**Figure 6.** Ovaroid follicle formation. (**A**) Day 35 human ovaroid (F66/N.R1.G #7 + hPGCLC) sections stained for FOXL2, OCT4, and AMHR2. Scale bars are 40 μm. Follicle-like structures are marked with yellow triangles. (**B**) Whole-ovaroid view of follicle-like structures in human ovaroids (F66/N.R1.G #7). Scale bars are 1 mm. (**C**) Section of human ovaroid (F66/N.R1.G.F #4 + hPGCLC) at day 70 of culture, stained for FOXL2, NR2F2, and AMHR2, showing multiple small follicles (yellow triangles) consisting single layers of FOXL2+AMHR2+ cells. NR2F2+ cells are interspersed between these. Scale bars are 100 μm. (**D**) Section of human ovaroid (F66/N.R2 #1 + hPGCLC) at day 70 of culture, stained for FOXL2, NR2F2, and AMHR2, showing an antral follicle consisting of FOXL2+AMHR2+ granulosa-like cells arranged in several layers around a central cavity. NR2F2 staining is visible outside of the follicle (marked 'Stroma'). Scale bars are 100 μm.

to exclusively cytoplasmic localization that was reported to take place at this stage. Expression of TFAP2C, an early-stage PGC marker, declined during ovaroid culture (*Figure 5D*) and was almost entirely absent by day 8. By contrast, SOX17 expression was still visible on day 8, and OCT4 and DAZL expression continued to day 54 (*Figure 5A, C*).

Although this system allowed the rapid development of hPGCLCs to the gonadal stage, the number of germ cells in both hiPSC- and mouse-derived ovaroids declined over prolonged culture (*Figure 5C*), indicating that either they were dying or differentiating to other lineages. Unlike mouse-derived ovaroids, the hiPSC-derived ovaroids cultured on Transwells gradually flattened and widened, and by day 38 were largely collapsed.

Nonetheless, in these longer-term experiments, we observed the formation of empty follicle-like structures composed of cuboidal AMHR2+FOXL2+ granulosa-like cells (*Figure 6A*), suggesting that the TFs could drive folliculogenesis even in the absence of oocytes. Follicle-like structure formation was first visible at day 16 (*Figure 5—figure supplement 1*), and at day 26 the largest of these structures had grown to 1–2 mm diameter (*Figure 6B*). At day 70, ovaroids had developed follicles of a variety of sizes, mainly small single-layer follicles (*Figure 6C*) but also including antral follicles (*Figure 6D*). Cells outside of the follicles stained positive for NR2F2 (*Figure 6C, D*), a marker of ovarian stromal and theca cells.

To further examine the gene expression of hPGCLCs and somatic cells in this system, we performed scRNA-seq on dissociated ovaroids at days 2, 4, 8, and 14 of culture, and clustered cells according to gene expression. As expected, the largest cluster (cluster 0) contained cells expressing granulosa markers such as *FOXL2*, *WNT4*, and *CD82* (*Figure 7A, B*). Cells expressing markers of secondary/antral granulosa cells such as *FSHR* and *CYP19A1* were also found within this cluster, although these were much less numerous. A smaller cluster (cluster 1) expressing the ovarian stromal marker *NR2F2* was also present. *NR2F2* is expressed by both stromal and theca cells, but the cells in cluster 1 did not express 17α-hydroxylase (*CYP17A1*), indicating that they could not produce androgens and were not theca cells. Full expression data for each cluster are given in *Figure 7—source data 1*.

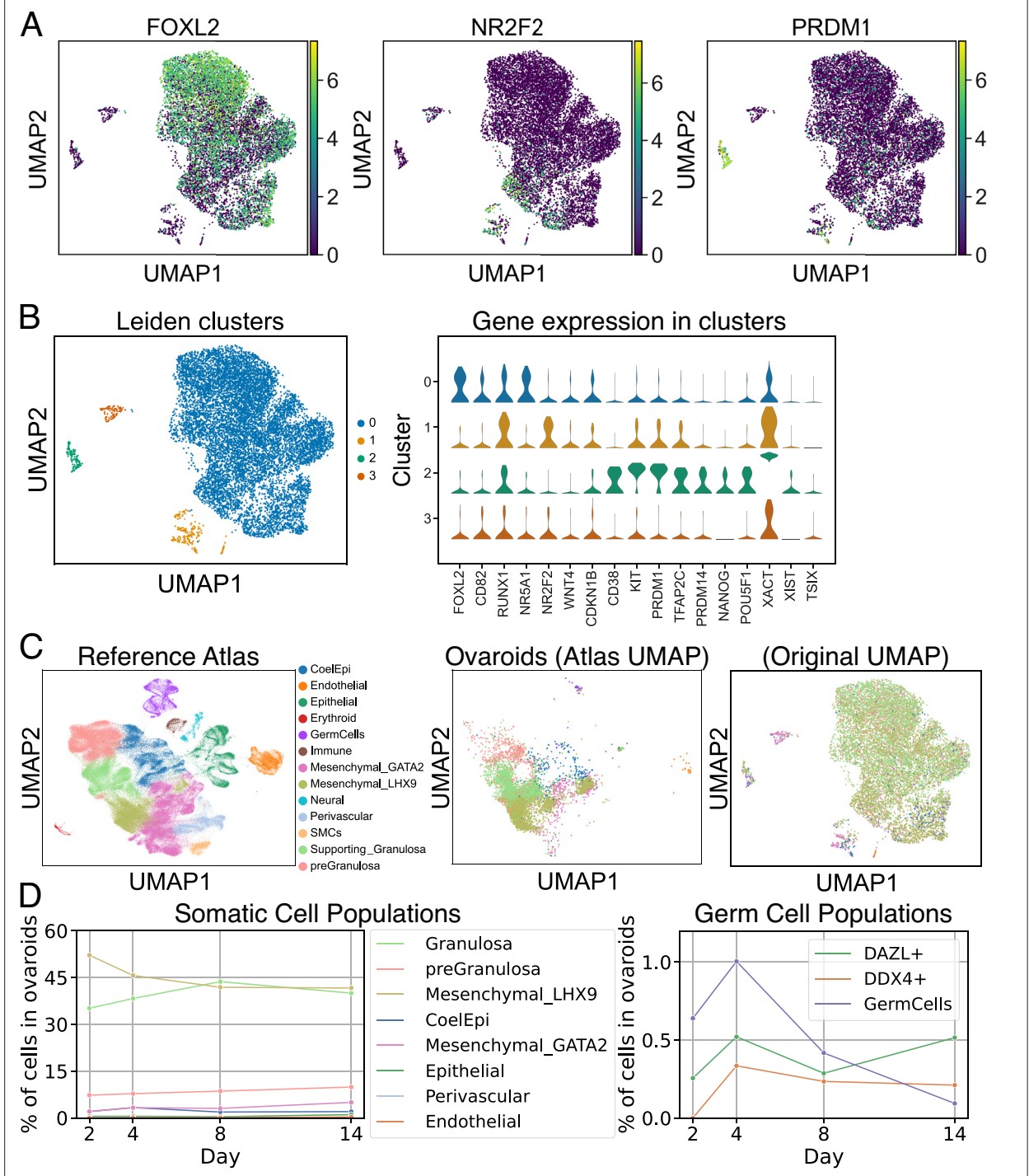

**Figure 7.** scRNA-seq analysis of ovaroids (F66/N.R1.G.F #4 granulosa-like cells + hPGCLCs). Data from all samples (days 2, 4, 8, and 14) were combined for joint dimensionality reduction and clustering. (**A**) Expression (log$_2$ CPM) of selected granulosa (*FOXL2*), stroma/theca (*NR2F2*), and germ cell (*PRDM1*) markers. (**B**) Leiden clustering shows four main clusters; the expression (log$_2$ CPM) of marker genes is plotted for each. (**C**) Mapping of cells onto a human fetal ovary reference atlas (***Garcia-Alonso et al., 2022***) and assignment of cell types. (**D**) Proportion of somatic cell types, germ cells, *DAZL+* cells, and *DDX4+* cells in ovaroids from each day.

The online version of this article includes the following source data for figure 7:

**Source data 1.** Cell populations from scRNA-seq and differential gene expression.

We also observed a cluster of hPGCLCs expressing marker genes such as *CD38*, *KIT*, *PRDM1*, *TFAP2C*, *PRDM14*, *NANOG*, and *POU5F1*. Notably, X-chromosomal lncRNAs *XIST*, *TSIX*, and *XACT* were all more highly expressed (an average of ~80-, ~20-, and ~2900-fold, respectively) in the hPGCLCs relative to other clusters (*Figure 7B*), suggesting that the hPGCLCs were starting the process of X-reactivation (*Vallot et al., 2017*), which in hPGCs is associated with high expression of both *XIST* and *XACT* (*Chitiashvili et al., 2020*). The X-chromosomal *HPRT1* gene, known to be more highly expressed in cells with two active X chromosomes (*Epstein, 1972*), was also ~3-fold upregulated (*Figure 7—source data 1*).

We next compared our in vitro-generated ovaroids to a reference atlas of human fetal ovarian development (*Garcia-Alonso et al., 2022*). We used scanpy ingest (*Wolf et al., 2018*) to integrate our samples into the atlas and annotate each cell with the closest cell type from the in vivo data (*Figure 7C*). The ovaroids consisted mainly of granulosa, gonadal mesenchyme, and pre-granulosa lineages (*Figure 7D*), with a small fraction of coelomic epithelium. The fraction of granulosa cells increased from day 2 through day 8, potentially representing a maturation of the somatic cell population. As expected, neural, immune, smooth muscle, and erythroid cells, which were present in fetal ovaries, were completely absent from our ovaroids. Epithelial, endothelial, and perivascular cells were detected, but at very low frequency (1% or less), possibly representing a low rate of off-target differentiation.

We additionally examined the overall fraction of germ cells, as well as the fraction of cells expressing the gonadal germ cell markers *DAZL* and *DDX4*, over the course of our experiment (*Figure 7D*). We defined the germ cell population based on the fetal ovary atlas integration. This population increased from days 2 to 4, but declined thereafter. In comparison, the fraction of *DAZL+* and *DDX4+* cells also increased from days 2 to 4, but remained roughly constant from days 4 to 14 (*Figure 7D*). We performed a differential gene expression analysis and gene ontology enrichment on *DAZL+* cells relative to *DAZL−* cells. Upregulated genes (log$_2$fc >2, $n$ = 221) were most highly enriched for terms related to generic developmental processes but also included terms related to adhesion and migration (e.g., 'ameboidal-type cell migration'), as well as reproductive system development. Downregulated genes (log$_2$fc <−2, $n$ = 6451) were strongly related to metabolic processes and mitotic cell division. Full data on differentially expressed genes and ontology enrichment are provided in *Supplementary file 4*. These data suggest that DAZL+ cells in our ovaroids are downregulating their metabolism and proliferation, in agreement with the known role of DAZL in suppressing PGC proliferation (*Yan et al., 2022*; *Gill et al., 2011*).

## Discussion

This work establishes a rapid (5 days) and efficient (>70% in top monoclonal lines) method to produce granulosa-like cells by TF-mediated differentiation of hiPSCs. These granulosa-like cells express granulosa marker genes and have highly similar transcriptomes to the human fetal ovary, both by bulk RNA-seq (*Figure 3*) and scRNA-seq (*Figure 7*). We demonstrate that our system can model several key ovarian phenotypes, including hormonal signaling, germ cell maturation, and follicle formation. Additionally, we characterize the regulatory effects of granulosa-related TFs and determine that co-expression of NR5A1 and RUNX1 or RUNX2 is minimally sufficient to generate granulosa-like cells from hiPSCs (*Figure 2B*). We show that FOXL2 and GATA4 may help to further stabilize the granulosa-like transcriptome (*Figure 3C*), and these TFs were also included in some of our top-performing lines (*Figure 3—figure supplement 1*, *Figure 3—source data 2*). Finally, our ovaroids formed from granulosa-like cells and hPGCLCs are notable as the first reported fully human ovarian organoids.

Our human ovaroids allow the rapid development of hPGCLCs to the gonadal stage as measured by expression of the markers *DAZL* and *DDX4* (*Figures 5 and 7*), which was previously only achieved by extended culture with mouse fetal gonadal somatic cells (*Yamashiro et al., 2018*; *Hwang et al., 2020*; *Murase et al., 2020*). As *DAZL* and *DDX4* are not expressed in pre-gonadal germ cells and only become expressed after germ cells colonize the gonads (*Yamashiro et al., 2018*; *Nicholls et al., 2019*; *Kobayashi et al., 2022*), this implies that our hiPSC-derived granulosa-like cells are able to simulate the gonadal niche and allow germ cell maturation. In a direct comparison with the previous method (*Figure 5*), we show that DAZL expression occurs earlier when hPGCLCs are co-cultured with hiPSC-derived granulosa-like cells instead of mouse fetal ovarian somatic cells.

While our results represent a step forward for the in vitro study of granulosa cells and germ cell development, future research will be necessary to construct a full model of the human ovary. Although we can efficiently generate granulosa-like cells and form ovaroids in combination with germ cells, improved methods will be required for maintaining these ovaroids in culture and preventing the loss of DAZL+ germ cells. Recent optimizations in media and culture methods for re-aggregated human fetal ovarian cells (*Mizuta et al., 2022*), which were specifically developed to avoid the 'flattening' issue with Transwell cultures, may be applicable to our system and allow for more extended development. Our results were also similar to a previous study in which rhesus PGCLCs gained expression of the mature marker DDX4 upon transplantation into an adult testis, but did not complete spermatogenesis (*Sosa et al., 2018*). A better developmental match (using later-stage germ cells or earlier-stage gonadal somatic cells) may provide better results.

Additionally, the ovary consists of more cell types than merely germ cells and granulosa cells (*Figure 7C*), and these other cells are also functionally important. For example, our ovaroids produced estradiol only in the presence of androstenedione, which is normally produced by ovarian theca cells. Although our granulosa-like cells were capable of a limited amount of differentiation into NR2F2+ ovarian stromal cells (*Figures 6 and 7*), these were not functional theca cells. Thus, developing a method to generate theca cells (or earlier-stage gonadal cells that differentiate into both granulosa and theca cells) will provide a more complete model of ovarian biology. We anticipate that our overall approach of identifying TFs to drive the differentiation of hiPSCs will be broadly applicable to generate these cell types of interest, as well as many others.

One further limitation is the random integration of our transposon-based TF expression vectors. This necessitated generating and testing several clonal hiPSC lines before selecting ones that performed well (*Figure 3—figure supplement 1* and *Figure 3—source data 2*). Additionally, due to the challenge of growing large numbers of hPGCLCs, we only performed our ovaroid experiments using granulosa-like cells from a limited number of lines (six for hormone assays, two for immunofluorescence, and one for scRNA-seq). We selected the lines for these endpoints based on an assessment of their surface marker expression and estradiol production, as well as preliminary aggregation experiments in which they showed follicle-like morphology. In future studies, we hope to better characterize the relative dosage requirements for our TFs and generate hiPSC lines with controlled TF expression from a known integration site in a safe-harbor locus, or from non-integrating episomal vectors. This should provide a more reproducible way to generate high-performing lines.

Despite these limitations, our system has several advantages in comparison with existing in vitro models of the human ovary. Several studies have used mouse fetal gonad cells to support hPGCLC development (*Yamashiro et al., 2018*; *Kobayashi et al., 2022*), but this process is much slower (32 vs. 4 days, *Figure 3*), the mouse cells may not behave the same as human cells, and mouse fetal gonad cells are difficult to obtain in large quantities. Other studies have used cancer cell lines such as COV434 or KGN to study various aspects of granulosa cells (*Price et al., 2012*; *Zhang et al., 2000*; *Nishi et al., 2001*). However, these cancer cells may be phenotypically aberrant. In particular, we show that unlike KGN cells or our iPSC-derived granulosa-like cells, COV434 cells do not express granulosa marker genes and do not respond to FSH. This is in agreement with a recent study reporting that the COV434 cell line was mis-identified and actually originates from small cell carcinoma of the ovary, not a granulosa tumor (*Karnezis et al., 2021*). Finally, some previous studies have reported growth-factor-based protocols to differentiate human pluripotent stem cells to ovarian granulosa-like cells (*Lipskind et al., 2018*; *Lan et al., 2013*; *Gonen et al., 2021*). However, the efficiency of such protocols is low (4–12%, or not reported), the protocols require extended culture (multiple weeks vs. 5 days), and those studies did not demonstrate that the cells could support germ cell maturation. Using monoclonal hiPSC lines, our induction efficiency (>70% in top lines, *Figure 3—figure supplement 1*) is sufficient to form ovaroids without the need for sorting FOXL2+ cells.

Overall, our rapid, scalable, and efficient method of producing human granulosa-like cells represents a substantial advance over previous studies. We anticipate that this system will prove useful

as a model for the study of ovarian development and function, and that further optimization could allow for human in vitro oogenesis.

# Materials and methods

## Key resources table

| Reagent type (species) or resource | Designation | Source or reference | Identifiers | Additional information |
|---|---|---|---|---|
| Strain, strain background (*Mus musculus*, female) | CD-1 | Charles River Labs | RRID:IMSR_CRL:022 | Used for fetal ovary isolation |
| Strain, strain background (*Mus musculus*, female) | BALB/c | Charles River Labs | RRID:IMSR_APB:4790 | Used for adult ovary isolation |
| Cell line (*Homo sapiens*, female) | F3 iPSC | ATCC | ATCC-BXS0116 | |
| Cell line (*Homo sapiens*, female) | F66 iPSC | Other | | Previously derived in-house from the NIA Aging Cell Repository fibroblast line AG07141 using Epi5 footprint-free episomal reprogramming |
| Cell line (*Homo sapiens*, female) | F2 PGCLC | *Kobayashi et al., 2022*; PMID:35148847 | PMID:35148847 | Derived from the ATCC-BXS0115 hiPSC line |
| Recombinant DNA reagent | pHDR-FOXL2-T2A-tdTomato-PuroTK | This paper | RRID:Addgene_192892 | For engineering reporter hiPSCs |
| Recombinant DNA reagent | pX330-FOXL2-cterm | This paper | RRID:Addgene_192893 | For engineering reporter hiPSCs |
| Recombinant DNA reagent | Super PiggyBac Transposase Expression Vector | System Bioscience | Cat:PB210PA-1 | |
| Recombinant DNA reagent | pPB-cT3G-cERP2-TCF21 | This paper | RRID:Addgene_192894 | Dox-inducible TF expression |
| Recombinant DNA reagent | pPB-cT3G-cERP2-JUN | This paper | RRID:Addgene_192895 | Dox-inducible TF expression |
| Recombinant DNA reagent | pPB-cT3G-cERP2-GATA4 | This paper | RRID:Addgene_192896 | Dox-inducible TF expression |
| Recombinant DNA reagent | pPB-cT3G-cERP2-CEBPD | This paper | RRID:Addgene_192897 | Dox-inducible TF expression |
| Recombinant DNA reagent | pPB-cT3G-cERP2-EMX2 | This paper | RRID:Addgene_192898 | Dox-inducible TF expression |
| Recombinant DNA reagent | pPB-cT3G-cERP2-MYC | This paper | RRID:Addgene_192899 | Dox-inducible TF expression |
| Recombinant DNA reagent | pPB-cT3G-cERP2-NR2F2 | This paper | RRID:Addgene_192900 | Dox-inducible TF expression |
| Recombinant DNA reagent | pPB-cT3G-cERP2-FOSB | This paper | RRID:Addgene_192901 | Dox-inducible TF expression |
| Recombinant DNA reagent | pPB-cT3G-cERP2-KLF2 | This paper | RRID:Addgene_192902 | Dox-inducible TF expression |
| Recombinant DNA reagent | pPB-cT3G-cERP2-ZBTB16 | This paper | RRID:Addgene_192903 | Dox-inducible TF expression |
| Recombinant DNA reagent | pPB-cT3G-cERP2-YBX1 | This paper | RRID:Addgene_192904 | Dox-inducible TF expression |

*Continued on next page*

*Continued*

| Reagent type (species) or resource | Designation | Source or reference | Identifiers | Additional information |
|---|---|---|---|---|
| Recombinant DNA reagent | pPB-cT3G-cERP2-LHX1 | This paper | RRID:Addgene_192905 | Dox-inducible TF expression |
| Recombinant DNA reagent | pPB-cT3G-cERP2-WT1 | This paper | RRID:Addgene_192906 | Dox-inducible TF expression |
| Recombinant DNA reagent | pPB-cT3G-cERP2-JUNB | This paper | RRID:Addgene_192907 | Dox-inducible TF expression |
| Recombinant DNA reagent | pPB-cT3G-cERP2-KLF4 | This paper | RRID:Addgene_192908 | Dox-inducible TF expression |
| Recombinant DNA reagent | pPB-cT3G-cERP2-KLF6 | This paper | RRID:Addgene_192909 | Dox-inducible TF expression |
| Recombinant DNA reagent | pPB-cT3G-cERP2-HOXC9 | This paper | RRID:Addgene_192910 | Dox-inducible TF expression |
| Recombinant DNA reagent | pPB-cT3G-cERP2-ATF4 | This paper | RRID:Addgene_192911 | Dox-inducible TF expression |
| Recombinant DNA reagent | pPB-cT3G-cERP2-PPARG | This paper | RRID:Addgene_192912 | Dox-inducible TF expression |
| Recombinant DNA reagent | pPB-cT3G-cERP2-EGR1 | This paper | RRID:Addgene_192913 | Dox-inducible TF expression |
| Recombinant DNA reagent | pPB-cT3G-cERP2-FOXL2 | This paper | RRID:Addgene_192914 | Dox-inducible TF expression |
| Recombinant DNA reagent | pPB-cT3G-cERP2-NR5A1 | This paper | RRID:Addgene_192915 | Dox-inducible TF expression |
| Recombinant DNA reagent | pPB-cT3G-cERP2-TSC22D3 | This paper | RRID:Addgene_192916 | Dox-inducible TF expression |
| Recombinant DNA reagent | pPB-cT3G-cERP2-MAFF | This paper | RRID:Addgene_192917 | Dox-inducible TF expression |
| Recombinant DNA reagent | pPB-cT3G-cERP2-ELK1 | This paper | RRID:Addgene_192918 | Dox-inducible TF expression |
| Recombinant DNA reagent | pPB-cT3G-cERP2-NR1H2 | This paper | RRID:Addgene_192919 | Dox-inducible TF expression |
| Recombinant DNA reagent | pPB-cT3G-cERP2-TOX2 | This paper | RRID:Addgene_192920 | Dox-inducible TF expression |
| Recombinant DNA reagent | pPB-cT3G-cERP2-LHX9 | This paper | RRID:Addgene_192921 | Dox-inducible TF expression |
| Recombinant DNA reagent | pPB-cT3G-cERP2-ZFPM2 | This paper | RRID:Addgene_192922 | Dox-inducible TF expression |
| Recombinant DNA reagent | pPB-cT3G-cERP2-HOPX | This paper | RRID:Addgene_192923 | Dox-inducible TF expression |
| Recombinant DNA reagent | pPB-cT3G-cERP2-FOS | This paper | RRID:Addgene_192924 | Dox-inducible TF expression |
| Recombinant DNA reagent | pPB-cT3G-cERP2-NR4A1 | This paper | RRID:Addgene_192925 | Dox-inducible TF expression |
| Recombinant DNA reagent | pPB-cT3G-cERP2-RUNX2 | This paper | RRID:Addgene_192926 | Dox-inducible TF expression |
| Recombinant DNA reagent | pPB-cT3G-cERP2-TAF4B | This paper | RRID:Addgene_192927 | Dox-inducible TF expression |
| Recombinant DNA reagent | pPB-cT3G-cERP2-RUNX1 | This paper | RRID:Addgene_192928 | Dox-inducible TF expression |
| Antibody | anti-AMHR2 (Rabbit IgG, polyclonal) | Thermo Fisher | Cat:PA5-13902; RRID:AB_2305463 | IF (1:100) |

*Continued on next page*

*Continued*

| Reagent type (species) or resource | Designation | Source or reference | Identifiers | Additional information |
|---|---|---|---|---|
| Antibody | anti-DAZL (Rabbit IgG, monoclonal) | Abcam | Cat:ab215718; RRID:AB_2893177 | IF (1:500) |
| Antibody | anti-FOXL2 (Goat IgG, polyclonal) | Novus | Cat:NB100-1277; RRID:AB_2106187 | IF (1:250) |
| Antibody | anti-44838 (Mouse IgG, monoclonal) | BD Bioscience | Cat:611202; RRID:AB_398736 | IF (1:200) |
| Antibody | anti-SOX17 (Goat IgG, polyclonal) | Novus | Cat:AF1924; RRID:AB_355060 | IF (1:500) |
| Antibody | anti-TFAP2C (Mouse IgG, monoclonal) | Abcam | Cat:ab110635; RRID:AB_10858471 | IF (1:250) |
| Antibody | anti-NR2F2 (Mouse IgG, monoclonal) | Novus | Cat:H7147; RRID:AB_1964214 | IF (1:250) |
| Antibody | anti-AMHR2-FITC (Rabbit IgG polyclonal) | Biorbyt | Cat:orb37457; RRID:AB_10992015 | Flow cytometry (1:60) |
| Antibody | anti-CD82-PerCP-Cy5.5 (Mouse IgG monoclonal) | BioLegend | Cat:342111; RRID:AB_2750124 | Flow cytometry (1:60) |
| Antibody | anti-EpCAM-APC-Cy7 (Mouse IgG monoclonal) | BioLegend | Cat:324245; RRID:AB_2783193 | Flow cytometry (1:60) |
| Antibody | anti-FSHR-APC (Mouse IgG monoclonal) | R&D Systems | Cat:FAB65591A; RRID:AB_2920602 | Flow cytometry (1:60) |
| Antibody | anti-Mouse IgG-AF647 (Donkey IgG polyclonal) | Fisher | Cat:A31571; RRID:AB_162542 | IF secondary (1:250) |
| Antibody | anti-Goat IgG-AF568 (Donkey IgG polyclonal) | Fisher | Cat:A11057; RRID:AB_2534104 | IF secondary (1:500) |
| Antibody | anti-Rabbit IgG-AF488 (Donkey IgG F(ab')2 polyclonal) | Jackson | Cat:711-546-152; RRID:AB_2340619 | IF secondary (1:500) |
| Commercial assay or kit | Estradiol ELISA | DRG | Cat:EIA-2693 | |
| Commercial assay or kit | Progesterone ELISA | DRG | Cat:EIA-1561 | |
| Commercial assay or kit | Evercode WT Mega scRNA-seq kit (v1) | Parse Biosciences | Cat:EC-W01050 | |
| Software, algorithm | Python scripts for data analysis | This paper (*Brixi et al., 2023*) | | https://github.com/programmablebio/granulosa, (copy archived at swh:1:rev:3c650290779db376c4d1f3a14960b08b17ae5561) |
| Software, algorithm | Parse Biosciences barcode alignment pipeline | Parse Biosciences | Version 0.9.6 | |

## Cell culture

Two parental hiPSC lines were used in this study: ATCC-BXS0116 female hiPSCs, which we refer to as the F3 line, and the F66 line, an in-house hiPSC line derived from the NIA Aging Cell Repository fibroblast line AG07141 using Epi5 footprint-free episomal reprogramming. The karyotypes of parental lines, as well as engineered reporter lines, were verified by Thermo Fisher Cell ID (single-nucleotide

polymorphism-based authentication) + Karyostat, and pluripotency was assessed by Thermo Fisher Pluritest. All lines were identified as normal.

hiPSCs were cultured in mTESR Plus medium (Stemcell Technologies) on standard polystyrene plates coated with hESC-qualified Matrigel (Corning). Medium was changed daily. Passaging was performed using 0.5 mM ethylenediaminetetraacetic acid solution in phosphate-buffered saline (PBS), or TRYPLE for experiments requiring single-cell dissociation. hiPSCs were treated with 10 μM Y-27632 (Ambeed) for 24 hr after each passage. COV434 cells were cultured in Dulbecco's modified Eagle medium (DMEM) + 10% fetal bovine serum (FBS) + 1× GlutaMax (Gibco). KGN cells (RIKEN, RCB1154) were cultured in DMEM/F12 + 10% FBS + 1× GlutaMax (Gibco). HGL5 cells (ABM cat. T0650) were cultured in Prigrow IV medium (ABM) with 10% FBS. Passaging was performed with TRYPLE (Gibco). hPGCLCs were cultured in S-CM medium as previously described (*Kobayashi et al., 2022*), and passaged with Accutase (Stemcell Technologies). Mycoplasma testing was performed by polymerase chain reaction (PCR) every 3 months; all cells tested negative.

## Electroporations

Electroporations were performed using a Lonza Nucleofector with 96-well shuttle, with 200,000 cells in 20 μl of P3 buffer. Pulse setting CA-137 was used for all electroporations. Selection with the appropriate agent was begun 48 hr after electroporation and continued for 5 days. For the agents used in this study, this time was sufficient to give a high-purity final cell population.

## Reporter construction

Homology arms for *FOXL2* were amplified by PCR from genomic DNA. A targeting plasmid, containing an in-frame C-terminal T2A-tdTomato reporter, as well as a Rox-PGK-PuroTK-Rox selection cassette (*Figure 2—figure supplement 1A*), was constructed by Gibson assembly. The plasmid backbone additionally had an MC1-DTA marker to select against random integration. sgRNA oligos targeting the C-terminal region of *FOXL2* were cloned into pX330 (Addgene #42230). For generation of the reporter lines, 1 μg donor plasmid and 1 μg sgRNA plasmid were co-electroporated into hiPSCs, which were subsequently plated in one well of a 6-well plate. After selection with puromycin (400 ng/ml), colonies were picked manually with a P20 pipette. The hiPSC lines generated were genotyped by PCR for the presence of wild-type and reporter alleles. Homozygous clones were further verified by PCR amplification of the entire *FOXL2* locus (*Figure 2—figure supplement 1B*) and Sanger sequencing.

To excise the selection cassette, hiPSCs were electroporated with pCAGGS-Dre (1 μg). Selection was performed with ganciclovir (4 μM) and colonies were picked as described above. The excision of the selection cassette was verified by genotyping. Primers used in this study are listed in *Supplementary file 1*.

## TF plasmid construction

TF cDNAs were obtained from the TFome (*Ng et al., 2021*) or the ORFeome (*ORFeome Collaboration, 2016*) as Gateway entry clones. These were cloned into a barcoded Dox-inducible expression vector (Addgene #175503) using MegaGate cloning (*Kramme et al., 2021b*). The final expression constructs were verified by Sanger sequencing, which also served to determine the barcode sequences for each TF. Two unique barcodes were used per TF during library pooling. Libraries were pooled using an equimolar quantity of each plasmid (measured using QuBit).

## TF screening for granulosa differentiation

A pooled library of barcoded TF plasmids was electroporated into FOXL2-tdTomato reporter hiPSCs, typically at 5 fmol library and 500 ng PiggyBac transposase expression plasmid (Systems Bio). These conditions were chosen to give an average copy number of approximately 5/cell (*Figure 2—figure supplement 2*). Some experiments were also performed at 50 fmol to explore the effects of higher copy numbers. For the screening data presented in *Figure 2*, two libraries were used: library #1, containing 35 TFs, and library #2, containing 18 TFs. Library #1 was used only at 5 fmol, whereas library #2 was used at both 5 and 50 fmol.

After selection with puromycin (400 ng/ml), hiPSCs were treated with doxycycline (1 μg/ml) in mTESR Plus medium. In additional experiments, hiPSCs were first differentiated to mesoderm following a previously published protocol (*Morizane and Bonventre, 2017*) before doxycycline

treatment. In both sets of experiments, doxycycline treatment continued for 5 days, after which the cells were dissociated with TRYPLE and reporter-positive cells were isolated by FACS. Genomic DNA was extracted (QIAamp DNA Micro kit) from reporter-positive and -negative cells, as well as from the initial population before doxycycline treatment.

Barcodes were amplified by PCR (KAPA polymerase), using 10 ng input gDNA per reaction and typically 22 PCR cycles (95°C 15 s denature, 58°C 20 s anneal/extend). PCR products were purified using ProNex beads, and a second round of PCR (NEB Q5 polymerase, six cycles of 98°C 5 s denature, 61°C 20 s anneal, 72°C 5 s extend, final extension 72°C 2 min) was performed to add Illumina indices (primers are given in *Supplementary file 1*). These amplicons were again purified using ProNex beads. Samples were normalized and pooled, and barcodes were sequenced on an Illumina MiSeq with 10% PhiX spike-in. To call barcodes, reads were aligned to the set of known barcode sequences. Fold changes were calculated by comparing barcode frequencies in the sorted FOXL2+ cells to the frequencies in the starting population.

## Flow cytometry/cell sorting

Cells were dissociated by treatment with TRYPLE for 5 min, which was quenched with 4 volumes of ice-cold DMEM + 10% FBS. The suspension was passed through a 70-µm cell strainer. Cells were pelleted (200 g, 5 min) and resuspended in staining buffer (PBS + 3% FBS + antibodies, approx. 100 µl per million cells). Staining continued on ice in the dark for 30 min. The suspension was diluted with 9 volumes of PBS + 3% FBS. Cells were pelleted (200 g, 5 min) and resuspended in PBS + 3% FBS + 100 ng/ml 4',6-diamidino-2-phenylindole (DAPI). The suspension was kept on ice in the dark until analysis. Flow cytometry was performed on a BD LSRFortessa, and sorting was performed on a Sony SH800 with 100-µm chip.

Antibody capture beads (BD Biosciences, RRID AB_10051478), or hiPSCs expressing tdTomato, were used as compensation controls. Antibodies used are given in the Key Resources table. Data analysis was performed using the Cytoflow Python package (version 1.0.0, https://github.com/cytoflow/cytoflow, *Teague, 2022*).

## Optimized protocol for granulosa differentiation

iPSCs were dissociated with TRYPLE, and plated in DK10 medium (DMEM–F12, 15 mM 4-(2-hydroxyethyl)-1-piperazineethanesulfonic acid [HEPES], 1× GlutaMax, 10% knockout serum replacement [KSR]) with Y-27632 (10 µM), CHIR99021 (3 µM), and doxycycline (1 µg/ml) at a cell density of 12,500/cm$^2$ on Matrigel-coated polystyrene plates. For 24-well plates the medium volume per well was 0.5 ml; for 6-well plates it was 2 ml. 48 hr after plating, the medium was changed to DK10 + doxycycline (1 µg/ml), and the medium was subsequently changed every 24 hr. Cells were harvested on day 5 unless otherwise indicated. In the no-TF control differentiation for RNA-seq, the protocol was the same except the cells did not contain TF expression plasmids.

## RNA-seq

Total RNA was extracted from sorted FOXL2+ granulosa-like cells using the Arcturus PicoPure kit (Thermo Fisher), or from COV434 cells and hiPSCs using the Monarch Total RNA Miniprep kit (NEB). For experiments involving TF overexpression, TF expression plasmids were integrated into hiPSCs as described above (50 fmol/200,000 cells). After selection with puromycin, TF expression was induced using doxycycline (1000 ng/ml).

Two biological replicates were collected for each sample (iPSC, hiPSC+ individual TFs, sorted FOXL2+, no-TF differentiation, KGN, COV434). Libraries were prepared using the NEBNext Ultra II Directional kit following the manufacturer's protocol, and sequenced on an Illumina NextSeq 500 (2 × 75 bp paired-end reads). The TPM data shown in *Figure 3* were generated using kallisto (*Bray et al., 2016*) to pseudoalign reads to the reference human transcriptome (Ensembl GRCh38 v96). Differential expression analysis was performed using DESeq2. PantherDB (*Mi et al., 2021*) was used to calculate gene ontology enrichment for significantly upregulated ($\log_2$fc >3, $p_{adj}$ < 0.05) and downregulated ($\log_2$fc <−3, $p_{adj}$ < 0.05) genes for each sample relative to hiPSCs.

## TROM analysis

The TROM method was employed to identify associated genes that capture molecular characteristics of biological samples and subsequently comparing the biological samples by testing the overlap

of their associated genes (*Li et al., 2017*). TROM scores were calculated as the $-\log_{10}$(Bonferroni corrected p value of association) on a scale of 0–300. The TROM magnitude is positively correlated with similarity between two independent samples, with a standard threshold of 12 as a generally accepted indicator of significant similarity.

## Ovaroid formation with hPGCLCs and granulosa-like cells

F2 female hPGCLCs (see Key Resources table) were maintained in long-term culture as previously described (*Kobayashi et al., 2022*). Briefly, hPGCLCs were cultured on Matrigel in STO-conditioned medium (Glasgow Minimum Essential Medium [GMEM] with 13% KSR and 1× non-essential amino acids, sodium pyruvate, and GlutaMax, all from Gibco), supplemented with stem cell factor (SCF) (100 ng/ml, Peprotech), ascorbic acid (50 µg/ml, Gibco), and 2-mercaptoethanol (25 µM, Gibco). hPGCLCs were harvested with Accutase. To form ovaroids, granulosa-like cells were harvested with TRYPLE, counted, and mixed with F2 hPGCLCs. For hormone assays in *Figure 3*, we used granulosa-like cells from F3/N.R1 #6, F66/N.R1.G.F #4, F66/N.R1.G #7, F66/N.R2 #1, F66/N.R2 #5, and F66/N.R2.G #3. For immunofluorescence experiments in *Figures 4 and 5*, we used F66/N.R1.G.F #4 and F66/N.R2 #1. For scRNA-seq in *Figure 6*, we used F66/N.R1.G.F #4.

For each ovaroid, 100,000 granulosa-like cells and 10,000 hPGCLCs were added to each well of a 96-well U-bottom low-bind plate (Corning #7007) in 200 µl of GK15 medium (GMEM, 15% KSR, with 1× GlutaMax, sodium pyruvate, and non-essential amino acids) supplemented with 10 mM Y-27632, 0.1 mM 2-mercaptoethanol, 1 µg/ml doxycycline, 100 ng/ml SCF, and 50 µg/ml primocin. The plate was centrifuged (100 × *g*, 2 min) and incubated (37°C, 5% $CO_2$) for 2 days. Subsequently, ovaroids were transferred to Transwells (collagen-coated polytetrafluoroethylene, 3-µm pore size, 24-mm diameter, Corning #3492) for air–liquid interface culture with Alpha Minimum Essential Medium, 10% KSR, 55 µM 2-mercaptoethanol, 500 ng/ml doxycycline, and 50 µg/ml primocin. Typically five to six ovaroids were cultured on each 6-well Transwell. The medium (1.5 ml) was changed every 2 days.

## Ovaroid formation with hPGCLCs and mouse fetal ovarian somatic cells

Fetal ovarian somatic cells were isolated from E12.5 female embryos of CD-1 mice (Charles River) as described by *Yamashiro et al., 2020*. For each ovaroid, 50,000 fetal ovarian somatic cells and 5000 F2 hPGCLCs were combined. Ovaroids were cultured as described above. All mouse experiments were approved by the Harvard Medical School Institutional Animal Care and Use Committee (IACUC).

## Immunofluorescence

Ovaroids were washed with PBS and fixed with 1% paraformaldehyde solution in PBS overnight at 4°C. After another PBS wash, ovaroids were detached from the Transwell. In preparation for cryosectioning, ovaroids were transferred to 10% sucrose in PBS. After 24 hr at 4°C, the 10% sucrose solution was removed and replaced with 20% sucrose in PBS. After an additional 24 hr at 4°C, the ovaroids were embedded in OCT compound and stored at −80°C until sectioning.

The ovaroids were sectioned to 10 µm using a Leica CM3050S cryostat. Sections were transferred to Superfrost Plus slides, which were washed with PBS to remove OCT compound. The slides were washed with PBST (0.1% Triton X-100 in PBS) and sections were circled with a Pap pen. Slides were blocked for 30 min at room temp. with blocking buffer (1% bovine serum albumin and 5% normal donkey serum [Jackson ImmunoResearch, #017-000-121, lot #152961] in PBST). The blocking buffer was removed and replaced with a solution of primary antibodies in blocking buffer, and the slides were incubated overnight at 4°C. The antibody solution was removed and the slides were washed with PBST for 3 × 5 min. The slides were incubated with secondary antibody and DAPI solution in blocking buffer for 1 hr. at room temp. in the dark, followed by two 5-min washes with PBST and one wash with PBS. After staining, samples were mounted in Prolong Gold medium and covered with coverslips. Imaging was performed on a Leica SP5 confocal microscope. Antibodies used are given the Key Resources Table. Images were adjusted for brightness (and only for brightness) in ImageJ (version 2.9.0/1.53t), and cell counts for *Figure 4C* were performed manually by a researcher who was blinded to the species of the ovaroids.

## Single-cell RNA sequencing

Ovaroids (6 ovaroids per sample, 2 samples per time point) were dissociated using the Miltenyi Embryoid Body Dissociation Kit (Miltenyi #130-096-348). The cells were passed through a 40-µm strainer, fixed using the Parse Biosciences fixation kit, and stored at −80°C until all time points had been collected. Libraries were prepared using the Parse Biosciences WT Mega v1 kit generating libraries of an average of 450 bp. The ovaroids took up 8 of the 96 samples; the remaining kit capacity was used for other experiments. The libraries were sequenced on an Illumina NovaSeq 2 × 150 bp S4 flow cell using single index, 6 bp, libraries and a 5% PhiX spike-in. Data were demultiplexed into library fastq files and counts matrices were generated using Parse Bioscience's analysis pipeline (version 0.9.6). Downstream data processing, such as doublet filtering, dimensionality reduction, and clustering, was performed using Scanpy (version 1.8.2) (*Wolf et al., 2018*). For cell type assignment, the fetal ovarian dataset from the human reproductive cell atlas (*Garcia-Alonso et al., 2022*) was used as a reference for scanpy ingest.

## Collection of primary mouse ovarian somatic cells

Female BALB/c mice (age 10–12 weeks) were confirmed to be in proestrus by visual examination. Mice were killed by $CO_2$ exposure followed by cervical dislocation, and ovaries were removed by dissection. Ovaries were placed in HEPES-buffered DMEM/F12 with 0.1% bovine serum albumin (2 ovaries per 1.5-ml tube, with 500-µl medium) and mechanically disrupted by stabbing with forceps. The cell suspensions were strained through a 40-µm strainer to remove oocytes and clumps prior to culture for hormone assays.

## Steroid hormone assays

Androstenedione (500 ng/ml) was added to the medium on day 4 of granulosa differentiation. FSH (0.25 IU/ml, BioVision #4781-50 lot 5F07L47810) or forskolin (100 µM, Sigma-Aldrich) were also added as indicated. The total medium volume was 0.5 ml per well of 24-well plate. We performed these assays on each of the lines listed in *Figure 3—source data 2*. For controls using human cell lines (COV434, KGN, or HGL5) or mouse primary ovarian somatic cells, 75,000 cells were seeded per well. After 24 hr, the medium was analyzed for estradiol content by ELISA (DRG International, EIA-2693). Concentrations were calculated with a 4-parameter logistic curve fit using the data from the standards provided in the kit. Samples outside the range of the calibration curve were diluted and re-run.

For measuring hormone production in ovaroids, ovaroids were aggregated as described above. Androstenedione (500 ng/ml) and/or FSH (0.25 IU/ml) were added to the aggregation medium (total volume 200 µl per ovaroid). After 3 days of culture, the medium was removed and analyzed by ELISA for estradiol (DRG International, EIA-2693) and progesterone (DRG International, EIA-1561). Hormone concentrations were calculated as described above.

## Materials availability

Cell lines generated in this study are available for noncommercial use; contact the authors to negotiate a Material Transfer Agreement. Plasmids generated in this study are available via Addgene; see the Key Resources Table for accession numbers.

# Acknowledgements

This work was funded by the Synthetic Biology Platform at the Wyss Institute for Biologically Inspired Engineering. This work was additionally funded by the Gameto Sponsored Research Agreement at Harvard University and Colossal Sponsored Research Agreement at Harvard University. MPS was supported by the National Science Foundation Graduate Research Fellowship. We thank Dr. Amanda Graveline and Oliver Dodd for assistance with mouse care and euthanasia. The KGN cell line was provided by the RIKEN BRC through the National BioResource Project of the MEXT, Japan.

# Additional information

## Competing interests

Merrick D Pierson Smela: is listed as an inventor for US Provisional Application No. 63/326,640, entitled 'Methods and Compositions for Producing Granulosa-Like Cells'. Christian C Kramme: is listed as an inventor for US Provisional Application No. 63/326,640, entitled 'Methods and Compositions for Producing Granulosa-Like Cells'. CK is the VP of Cell Engineering of Gameto, Inc. Pranam Chatterjee: is listed an an inventor for US Provisional Application No. 63/326,640, entitled 'Methods and Compositions for Producing Granulosa-Like Cells'. PC is a co-founder and scientific advisor to Gameto, Inc. George M Church: is listed as an inventor for US Provisional Application No. 63/326,640, entitled 'Methods and Compositions for Producing Granulosa-Like Cells'. GMC serves on the scientific advisory board of Gameto, Inc, Colossal Biosciences, and GCTx. The other author declares that no competing interests exist.

## Funding

| Funder | Grant reference number | Author |
| --- | --- | --- |
| Wyss Institute | Synthetic Biology Platform | George M Church |
| Gameto, Inc | Sponsored Research Agreement | George M Church |
| National Science Foundation | Graduate Research Fellowship. | Merrick D Pierson Smela |

The funders had no role in study design, data collection, and interpretation, or the decision to submit the work for publication.

## Author contributions

Merrick D Pierson Smela, Conceptualization, Resources, Data curation, Software, Formal analysis, Validation, Investigation, Visualization, Methodology, Writing – original draft, Writing – review and editing; Christian C Kramme, Conceptualization, Resources, Investigation, Methodology; Patrick RJ Fortuna, Investigation, Visualization, Writing – review and editing; Jessica L Adams, Rui Su, Edward Dong, Investigation; Mutsumi Kobayashi, Resources; Garyk Brixi, Venkata Srikar Kavirayuni, Emma Tysinger, Software; Richie E Kohman, Project administration; Toshi Shioda, Resources, Supervision, Writing – review and editing; Pranam Chatterjee, Software, Project administration, Supervision, Writing – review and editing; George M Church, Supervision, Funding acquisition, Project administration, Writing – review and editing

## Author ORCIDs

Merrick D Pierson Smela ⓘ http://orcid.org/0000-0001-5816-7098
Christian C Kramme ⓘ http://orcid.org/0000-0002-7518-8111
Mutsumi Kobayashi ⓘ http://orcid.org/0000-0001-7897-2633
Emma Tysinger ⓘ http://orcid.org/0000-0002-3958-8097
Richie E Kohman ⓘ http://orcid.org/0000-0002-7412-671X
Toshi Shioda ⓘ http://orcid.org/0000-0002-9434-7835
Pranam Chatterjee ⓘ http://orcid.org/0000-0003-3957-8478
George M Church ⓘ http://orcid.org/0000-0003-3535-2076

## Ethics

All protocols involving mice were approved by the Institutional Animal Care and Use Committee (IACUC) of Harvard Medical School, with reference number IS00001087.

## Decision letter and Author response

Decision letter https://doi.org/10.7554/eLife.83291.sa1
Author response https://doi.org/10.7554/eLife.83291.sa2

## Additional files

**Supplementary files**
• Supplementary file 1. List of oligonucleotides. This file contains all the oligonucleotide sequences used for this project.
• Supplementary file 2. Media optimization for granulosa-like cell induction. This file contains the analyzed flow cytometry data from a preliminary optimization experiment, in which we performed our granulosa differentiation protocol on a polyclonal population of FOXL2-tdTomato reporter iPSCs with NR5A1 and RUNX1 expression vectors integrated. We tested three different basal media (DMEM–F12 + 10% KSR, aRPMI + B27, and mTeSR) and various additives (CHIR99021, FGF, TGF-beta, BMP4, and estradiol). We monitored expression of FOXL2-T2A-tdTomato, CD82, FSHR, and EpCAM.
• Supplementary file 3. Gene ontology term enrichment. A differential gene expression analysis was performed (relative to iPSC control) on transcription factor (TF) overexpression samples (*Figure 3*) as well as iPSCs differentiated according to our granulosa differentiation protocol, but in the absence of TF expression. These files list significantly (FDR <0.05) enriched gene ontology terms for biological processes among upregulated ($\log_2$fc >3, p < 0.05) and downregulated ($\log_2$fc <−3, p < 0.05) genes relative to iPSC control for each condition.
• Supplementary file 4. Differential gene expression in ovaroid scRNA-seq. This file contains lists of genes differentially expressed in each of the Leiden clusters, as well as a list of genes differentially expressed in *DAZL+* vs. *DAZL−* germ cells and enriched gene ontology terms.
• MDAR checklist

**Data availability**

All data needed to evaluate the conclusions in the paper are present in the paper and supplementary tables and figures. Data analysis code can be found at: https://github.com/programmablebio/granulosa, (copy archived at swh:1:rev:3c650290779db376c4d1f3a14960b08b17ae5561). Raw and processed sequencing data have been deposited to GEO under accession GSE213156.

The following dataset was generated:

| Author(s) | Year | Dataset title | Dataset URL | Database and Identifier |
|---|---|---|---|---|
| , Smela MP, Kramme C, Fortuna P, Adams J, Dong E, Kobayashi M, Brixi G, Tysinger E, Kohman RE, Shioda T, Chatterjee P, Church GM | 2022 | Directed Differentiation of Human iPSCs to Functional Ovarian Granulosa-Like Cells via Transcription Factor Overexpression | http://www.ncbi.nlm.nih.gov/geo/query/acc.cgi?acc=GSE213156 | NCBI Gene Expression Omnibus, GSE213156 |

The following previously published datasets were used:

| Author(s) | Year | Dataset title | Dataset URL | Database and Identifier |
|---|---|---|---|---|
| Irie N, Weinberger L, Tang WW, Kobayashi T, Viukov S, Manor Y, Dietmann S, Hanna JH, Surani MA | 2014 | SOX17 Is a Critical Specifier of Human Primordial Germ Cell Fate | https://www.ncbi.nlm.nih.gov/geo/query/acc.cgi?acc=GSE60138 | NCBI Gene Expression Omnibus, GSE60138 |
| Garcia-Alonso L, Lorenzi V | 2022 | Single-cell roadmap of human gonadal development | https://cellgeni.cog.sanger.ac.uk/vento/reproductivecellatlas/gonads/human_main_female.h5ad | Reproductive Cell Atlas VentoLab, human_main_female.h5ad |
| Lecluze E, Rolland AD, Chalmel F | 2020 | Dynamics of the transcriptional landscape during human gonad development during fetal life | https://www.ncbi.nlm.nih.gov/geo/query/acc.cgi?acc=GSE116278 | NCBI Gene Expression Omnibus, GSE116278 |

*Continued on next page*

*Continued*

| Author(s) | Year | Dataset title | Dataset URL | Database and Identifier |
|---|---|---|---|---|
| Yatsenko SA, Wood-Trageser M, Chu T, Jiang H | 2019 | Fetal ovary expression profile | https://www.ncbi.nlm.nih.gov/geo/query/acc.cgi?acc=GSE126893 | NCBI Gene Expression Omnibus, GSE126893 |

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
