## [Editor Report]

This manuscript addresses a fundamental issue in ovarian biology of deriving granulosa cells from human iPS cells. These findings are important to treat female infertility in the future and may prove valuable in the Ob and Gyn clinical practice. The authors provide compelling evidence by developing and validating their model using in vitro ovaroids. This study provides a novel resource for transcriptomic signatures of ovarian somatic cells derived in vitro.

---

## [Decision Letter]

**Decision letter after peer review:**

Thank you for submitting your article "Directed Differentiation of Human iPSCs to Functional Ovarian Granulosa-Like Cells via Transcription Factor Overexpression" for consideration by *eLife*. Your article has been reviewed by 3 peer reviewers, and the evaluation has been overseen by a Reviewing Editor and Marianne Bronner as the Senior Editor. The reviewers have opted to remain anonymous.

Essential revisions:

1) Include a diagram to illustrate the experimental workflow of the study

2) Improve the quality of images and colors in Figure 3 and 4. In Figure 3, the *Foxl2*+ cells do not seem to be in a position similar to what is typically observed in vivo in ovaries.

3) Provide a clear rationale for using different endpoints for different cell lines.

4) The ovaroid data requires an OCT4+ and DAZL+ expression over the total analysis of the study.

5) The actual transcription factors used in the study need to be clearly described in the Discussion section.

*Reviewer #1 (Recommendations for the authors):*

I got lost at the beginning of the manuscript trying to understand exactly all the steps that led to the generation of the granulosa-like cells. It would be helpful to include a diagram to illustrate the experimental workflow of the study similar to a graphical abstract.

In Figure 3, I have several recommendations to facilitate the interpretation of the data

– The authors mention several times that they observe follicle-like formation in their ovaroid model, but in the IF images the *FOXL2*+ and AMHR2 staining is scattered and looks nothing like the spherical arrangement of squamous (primordial) or cuboidal (growing) cells one would expect to see in the in vivo ovary. In addition, there is no evidence in figures 3 or S5 that DAZL+ hPGCLCs become surrounded by *FOXL2*+ granulosa-like cells. I suggest replacing the images with better-quality IF stains, or the authors should soften the conclusion that follicle-like structures do form in their ovaroid model.

– The images in panel A Day 2 and Day 4 appear to be on a different scale than the rest, which is confusing.

– The merged images on the right are not legible, consider removing the DAPI channel from these so that the reader can better appreciate the staining overlap or lack thereof.

– In figure 4C, I find the integration onto the human cell atlas very helpful. However, it is confusing that the UMAP no longer corresponds to the cluster distribution shown in figure 4A. I would suggest adding an additional visual that shows the projection of the human cell atlas cell ID onto the original clusters of the ovaroid scRNAseq. Also, how do the authors explain that RUNX1 expression is very low in the *FOXL2*+ granulosa-like cluster (Figure 4B)?

Some of the data is difficult to evaluate in the figures.

– The bar histograms in figures 2d and e are very compact and difficult to tell apart. Also, the colors are not colorblind-friendly. I recommend changing the colors to more accessible color combinations in these and in figure 4D.

– The images of the ovaroids lack annotations to guide the readers to what is relevant, and the staining quality is suboptimal, making it difficult to interpret the similarity of these ovariods to in vivo ovaries.

*Reviewer #2 (Recommendations for the authors):*

This study from Smela et al. describes the production and functional characterisation of granulosa-like cells from induced pluripotent stem cells. The key claim is that two transcription factors – NR5A1, and either RUNX1 or RUNX2 – are collectively necessary and sufficient for the production of these cells (abstract line 9, results page 7 line 21, discussion page 16 line 9). While sufficiency for a selected set of characteristics is in part supported by functional and transcriptional analyses, the claim of necessity is not directly tested. If this claim of necessity is to stand, then the authors must clarify their use of this term, or alternatively, describe experiments they have undertaken that would justify this claim. Given the extensive data presented in figure 1b, the threshold used for this justification should also be described in much greater detail.

Impressively, this paper describes the rapid development of the germline when human-human chimeras are used, and this is a major strength of the study (figure 3). I do however find the use of the term "ovaroid" to be somewhat problematic. An ovary contains many more than two cell types, and a more nuanced term is strongly advised, potentially related to the production of a follicle-like structure.

While many suitable approaches are performed, the use of COV434 cells is not well-founded. As the authors note, there is some uncertainty regarding the identity of COV434 cells, and as a cancer cell type, are unlikely to reflect embryonic granulosa cells. A more appropriate control might be the starting iPS cells (as presented), or the various sublines developed, from which a more realistic assessment of the TROM value can be drawn. As presented, this transcriptional analysis cannot be used to justify the identity of the TF-induced granulosa-like cells. The authors acknowledge the issues with COV434 cells on page 9 lines 11-14, which further justifies an alternative approach.

The regulatory logic described in Figure 2c is not clearly described, and a much more rigorous description of the data underlying these findings should be included.

The section that describing the functional maturation of germline cells (via activation of DAZL and DDX4) is limited in scope, and the manuscript would benefit from orthogonal approaches to demonstrate the engraftment of these cells into follicle-like structures, or transcriptional analyses supporting the claims.

*Reviewer #3 (Recommendations for the authors):*

Additional Analyses:

1. Are multiple iPSC lines – biological variables – used to differentiate into granulosa-like cells? It seems this was only tested on lines from ATCC-BXS0116 and not confirmed in another biologically different line. Testing additional lines with the established protocol is necessary to increase robustness.

2. There are no images of ovaroids treated with androstenedione or FSH for the E2 and P4 experiments.

3. The ovaroid data was descriptive and requires an OCT4+ and DAZL+ cell over total analysis overtime to support the descriptive claims that these cell types underwent "rapid development" and "declined over time" and to become compare the results from the hPSCs against the murine fetal cells. While the authors do perform an analysis of Dazl-expressing cells in Figure 4D, this is not what is referred to in the paragraphs of results above and there are no comparisons with murine ovaroids here.

4. Which cells in the ovaroid come from the granulosa-like cell lines versus those that were differentiated into PGCLCs? It will be important to distinguish these populations in "follicle-like" cultures and to confirm that both are still present.

Editorial or data description suggestions:

1. "The description of experimentation within the Results section must be more clear."

a. The data represented in Figure 2C and SFiles 2 must be presented in a form that demonstrates the relative fold change of expression for each doxy-inducible transcription factor lines in a main figure.

b. The description of iPSC lines that have one or more TF must be clearly stated. And the results must be clearly described for each line tested. For example, on page 11 "The fourth of our granulosa lines produced high levels of estradiol in all conditions, and this line lacked the RUNX1 expression vector that was present in the others." Which four lines were used and in which order the authors are referring to them to deduce the "fourth line" was not made clear. Which TFs were in the lines that were picked for "phenotype" on page 12? First described on page 13 – which lines were used as the granulosa-like lines?

c. Figure 2 – 2 biological replicates for 4 clones? What does G1, G2, G3, G4 mean in Figure 2E? Are these different monoclonal granulosa-like cells from those shown in 2D? If so, why were they chosen?

2. It is not obvious which iPSC lines were used to generate the hPGCLCs or GLCs in the ovaroid cultures. This should be stated in the results and methods.

3. Page 13 lines 12-14 compare the expression of SOX17, OCT4, DAZL, and TFAP2C. However, the day in culture chosen for these IF analyses were different. This sentence should be reworded to more accurately state that OCT4 and DAZL expression was still visible on day 32 and SOX17 expression on day 8 in human…

4. Are OCT4- and DAZL+ oogonia visible on day 32? The line on page 13, line 14 states that "at later timepoints (day 16), DaZL+…" This is confusing because there is an image for DAZL and OCT4 antibodies at day 32 in Figure 3.

5. The germ cell populations in Figure 4D are not mentioned in the text.

6. The discussion should clearly summarize/restate the transcription factors used and steps to efficiently generate the granulosa-like cells from hiPSCs.

7. There needs to be clarity on which lines were used for each experiment in the Materials and methods.

---

## [Author Response]

Essential revisions:1) Include a diagram to illustrate the experimental workflow of the study

We have made this diagram and included it as Figure 1.

2) Improve the quality of images and colors in Figure 3 and 4. In Figure 3, the Foxl2+ cells do not seem to be in a position similar to what is typically observed in vivo in ovaries.

We have changed our graphs to have a colorblind-friendly palette. Additionally, we have improved the brightness of our immunofluorescence images (previously we were only showing the raw brightness values), and exported the figures at a higher resolution.

We have also collected additional immunofluorescence images from day 70 of ovaroid culture, which clearly show follicle-like structures (Figure 6).

3) Provide a clear rationale for using different endpoints for different cell lines.

We have added this to the Discussion section. Also, we collected additional estradiol production and FSH response data, so that we now have that data for 9 cell lines instead of 4 as before.

4) The ovaroid data requires an OCT4+ and DAZL+ expression over the total analysis of the study.

We have included this as Figure 5C.

5) The actual transcription factors used in the study need to be clearly described in the Discussion section.

We have added this to the Discussion section, added a table showing which TFs are in which lines (Figure 3 – Source Data 2), and also provided PCR data verifying this (Figure 3—figure supplement 1C). Additionally, we have re-named our cell lines to be more informative about which TFs they express.

Reviewer #1 (Recommendations for the authors):I got lost at the beginning of the manuscript trying to understand exactly all the steps that led to the generation of the granulosa-like cells. It would be helpful to include a diagram to illustrate the experimental workflow of the study similar to a graphical abstract.

We have included this as Figure 1.

In Figure 3, I have several recommendations to facilitate the interpretation of the data– The authors mention several times that they observe follicle-like formation in their ovaroid model, but in the IF images the FOXL2+ and AMHR2 staining is scattered and looks nothing like the spherical arrangement of squamous (primordial) or cuboidal (growing) cells one would expect to see in the in vivo ovary. In addition, there is no evidence in figures 3 or S5 that DAZL+ hPGCLCs become surrounded by FOXL2+ granulosa-like cells. I suggest replacing the images with better-quality IF stains, or the authors should soften the conclusion that follicle-like structures do form in their ovaroid model.

We have collected additional immunofluorescence images from day 70 of ovaroid culture, which clearly show follicle-like structures of *FOXL2*+ AMHR2+ cells (Figure 5).

We agree that the follicles are empty (not containing hPGCLCs), and we mention this in the Results section.

– The images in panel A Day 2 and Day 4 appear to be on a different scale than the rest, which is confusing.

We have swapped these for different images which have the same magnification as the rest.

– The merged images on the right are not legible, consider removing the DAPI channel from these so that the reader can better appreciate the staining overlap or lack thereof.

We have done this, and also changed the colors to make them more legible.

– In figure 4C, I find the integration onto the human cell atlas very helpful. However, it is confusing that the UMAP no longer corresponds to the cluster distribution shown in figure 4A. I would suggest adding an additional visual that shows the projection of the human cell atlas cell ID onto the original clusters of the ovaroid scRNAseq.

We have done this.

Also, how do the authors explain that RUNX1 expression is very low in the FOXL2+ granulosa-like cluster (Figure 4B)?

*RUNX1* expression is not very low in the granulosa cluster. In the original version of the violin plot (which is now figure 7B), the default setting of the width scaling factor was too small, which made it look like there wasn’t much *RUNX1*. We have since corrected this. The distribution of *RUNX1* expression is shown in Author response image 1.

**Author response image 1. sa2fig1:** 

Some of the data is difficult to evaluate in the figures.– The bar histograms in figures 2d and e are very compact and difficult to tell apart. Also, the colors are not colorblind-friendly. I recommend changing the colors to more accessible color combinations in these and in figure 4D.

We have changed the colors in these figures to a colorblind-friendly palette. Also, we separated the ELISA and bulk RNA-seq figures so that the bar plots can be bigger.

– The images of the ovaroids lack annotations to guide the readers to what is relevant, and the staining quality is suboptimal, making it difficult to interpret the similarity of these ovariods to in vivo ovaries.

We have added annotations to better show the follicle-like structures in figure 6.

The figure files in the original manuscript were at lower resolution because there was a file size limit on the PDF upload. We have provided original-resolution versions of the figures in our resubmission. We note that the original versions of our immunofluorescence figures were simply the raw image files not adjusted for brightness/contrast. We have since adjusted the brightness of these images in the figures (as most other papers do), and this improves their interpretability. We have also chosen a better color palette. There is still some background staining in a few of the images but this should not affect the conclusions of our paper.

Reviewer #2 (Recommendations for the authors):This study from Smela et al. describes the production and functional characterisation of granulosa-like cells from induced pluripotent stem cells. The key claim is that two transcription factors – NR5A1, and either RUNX1 or RUNX2 – are collectively necessary and sufficient for the production of these cells (abstract line 9, results page 7 line 21, discussion page 16 line 9). While sufficiency for a selected set of characteristics is in part supported by functional and transcriptional analyses, the claim of necessity is not directly tested. If this claim of necessity is to stand, then the authors must clarify their use of this term, or alternatively, describe experiments they have undertaken that would justify this claim. Given the extensive data presented in figure 1b, the threshold used for this justification should also be described in much greater detail.

We have removed the claim of necessity, since it’s possible that other TFs outside of the set we tested could also produce granulosa-like cells.

Impressively, this paper describes the rapid development of the germline when human-human chimeras are used, and this is a major strength of the study (figure 3). I do however find the use of the term "ovaroid" to be somewhat problematic. An ovary contains many more than two cell types, and a more nuanced term is strongly advised, potentially related to the production of a follicle-like structure.

Although our organoids do not contain all the cell types of the ovary, they do contain several important ones such as *FOXL2*+ granulosa-like cells, NR2F2+ stroma-like cells, and hPGCLCs. The name “ovaroid” is intended to describe a system resembling an ovary, not an actual ovary.

While many suitable approaches are performed, the use of COV434 cells is not well-founded. As the authors note, there is some uncertainty regarding the identity of COV434 cells, and as a cancer cell type, are unlikely to reflect embryonic granulosa cells. A more appropriate control might be the starting iPS cells (as presented), or the various sublines developed, from which a more realistic assessment of the TROM value can be drawn. As presented, this transcriptional analysis cannot be used to justify the identity of the TF-induced granulosa-like cells. The authors acknowledge the issues with COV434 cells on page 9 lines 11-14, which further justifies an alternative approach.

We have collected additional RNA-seq data from KGN cells, which are phenotypically closer to bona fide granulosa cells in terms of steroidogenesis and marker gene expression. In our TROM analysis, the control is the starting iPSCs (which we find to have a TROM score of zero with all of the other samples).

The regulatory logic described in Figure 2c is not clearly described, and a much more rigorous description of the data underlying these findings should be included.

We have clarified that this chart displays upregulation of genes of interest in response to TF overexpression. We have also included volcano plots to numerically show the regulatory effects.

The section that describing the functional maturation of germline cells (via activation of DAZL and DDX4) is limited in scope, and the manuscript would benefit from orthogonal approaches to demonstrate the engraftment of these cells into follicle-like structures, or transcriptional analyses supporting the claims.

Although we now have strong evidence for the formation of follicle-like structures (Figure 6) we have not observed any germ cells inside of these follicles. We do mention that these follicles are empty in the Results section. Future studies using optimized culture conditions to promote germ cell survival will be necessary to address this. Regarding transcriptional analysis, we have now performed a differential expression and gene ontology enrichment analysis on *DAZL+* vs *DAZL-* germ cells in our ovaroids, which we describe in the Results section.

Reviewer #3 (Recommendations for the authors):Additional Analyses:1. Are multiple iPSC lines – biological variables – used to differentiate into granulosa-like cells? It seems this was only tested on lines from ATCC-BXS0116 and not confirmed in another biologically different line. Testing additional lines with the established protocol is necessary to increase robustness.

We tested this on both F3 (ATCC-BXS0116) and F66 (an hiPSC line derived in-house from NIA Aging Cell Repository fibroblast line AG07141). A full list of lines is given in Figure 3 – Source Data 2.

2. There are no images of ovaroids treated with androstenedione or FSH for the E2 and P4 experiments.

We tried to do whole-mount staining on these, but this was unsuccessful and we were not able to get any good images. We do have brightfield images of the ovaroids, but these are not very informative (the samples are too thick to see much detail), and there do not seem to be any major differences in appearance between the conditions.

3. The ovaroid data was descriptive and requires an OCT4+ and DAZL+ cell over total analysis overtime to support the descriptive claims that these cell types underwent "rapid development" and "declined over time" and to become compare the results from the hPSCs against the murine fetal cells. While the authors do perform an analysis of Dazl-expressing cells in Figure 4D, this is not what is referred to in the paragraphs of results above and there are no comparisons with murine ovaroids here.

We have performed this analysis and it is now presented in Figure 5C.

4. Which cells in the ovaroid come from the granulosa-like cell lines versus those that were differentiated into PGCLCs? It will be important to distinguish these populations in "follicle-like" cultures and to confirm that both are still present.

We do not currently have enough information to definitively assign the origins of the cells in the ovaroids. However, we consider it unlikely that DAZL+ cells could originate from our granulosa-like cells, since DAZL expression is highly specific to gonadal PGCs (see Refs. 15–17). Also we do not see *DAZL* expression in bulk RNA-seq data of our *FOXL2*+ cells (see Source Data for figure 3A).

Editorial or data description suggestions:1. "The description of experimentation within the Results section must be more clear."a. The data represented in Figure 2C and SFiles 2 must be presented in a form that demonstrates the relative fold change of expression for each doxy-inducible transcription factor lines in a main figure.

We have added volcano plots to Figure 3 which depict the fold-changes for genes of interest, including our transcription factors. Full data are provided in the source data for that figure.

b. The description of iPSC lines that have one or more TF must be clearly stated. And the results must be clearly described for each line tested. For example, on page 11 "The fourth of our granulosa lines produced high levels of estradiol in all conditions, and this line lacked the RUNX1 expression vector that was present in the others." Which four lines were used and in which order the authors are referring to them to deduce the "fourth line" was not made clear. Which TFs were in the lines that were picked for "phenotype" on page 12? First described on page 13 – which lines were used as the granulosa-like lines?

We changed our naming of the clones to clearly describe which TFs are present, and also made a list of all the clones in Figure 3 – Source Data 2.

c. Figure 2 – 2 biological replicates for 4 clones? What does G1, G2, G3, G4 mean in Figure 2E? Are these different monoclonal granulosa-like cells from those shown in 2D? If so, why were they chosen?

We have tested additional clones (now all nine of our top clones, instead of just four). We also changed our naming of the clones to be more descriptive.

2. It is not obvious which iPSC lines were used to generate the hPGCLCs or GLCs in the ovaroid cultures. This should be stated in the results and methods.

We have added this.

3. Page 13 lines 12-14 compare the expression of SOX17, OCT4, DAZL, and TFAP2C. However, the day in culture chosen for these IF analyses were different. This sentence should be reworded to more accurately state that OCT4 and DAZL expression was still visible on day 32 and SOX17 expression on day 8 in human…

We reworded this following your suggestion.

4. Are OCT4- and DAZL+ oogonia visible on day 32? The line on page 13, line 14 states that "at later timepoints (day 16), DaZL+…" This is confusing because there is an image for DAZL and OCT4 antibodies at day 32 in Figure 3.

We clarified that DAZL+/OCT4- and DAZL+/OCT4+ cells are both present.

5. The germ cell populations in Figure 4D are not mentioned in the text.

The “germ cells” are the cells annotated as germ cells by our fetal ovary atlas integration. We have indicated this in the text.

6. The discussion should clearly summarize/restate the transcription factors used and steps to efficiently generate the granulosa-like cells from hiPSCs.

We have added this.

7. There needs to be clarity on which lines were used for each experiment in the Materials and methods.

We have added this.